



# Extreme Flood Impact on Estuarine and Coastal Biogeochemistry: the 2013 Elbe Flood

Yoana G. Voynova[1], Holger Brix[1], Wilhelm Petersen[1], Sieglinde Weigelt-Krenz[2], Mirco Scharfe[3]

[1]Institute of Coastal Research, Helmholz-Zentrum Geesthacht (HZG), 21502 Geesthacht, Germany
[2]Federal Maritime and Hydrographic Agency, BSH-Laboratory Sülldorf, 22589 Hamburg, Germany

[3]Alfred Wegener Institut Helmholtz-Zentrum für Polar- und Meeresforschung, Biologische Anstalt Helgoland, P.O. Box 180, 27483 Helgoland, Germany

*Correspondence to*: Yoana G. Voynova (yoana.voynova@hzg.de)





**Abstract.** Within the context of predicted and observed increase in droughts and floods with climate change, large summer floods are likely to become more frequent. These extreme events can alter typical biogeochemical patterns in coastal systems. The extreme Elbe River flood in June, 2013 not only caused

major damages in several European countries, but also generated large scale biogeochemical changes in the Elbe Estuary and the adjacent German Bight. Due to a number of well documented and unusual atmospheric conditions, the early summer of 2013 in Central and Eastern Europe was colder and wetter than usual, with saturated soils, and higher than average cumulative precipitation. Additional precipitation at the end of May, and beginning of June, 2013, caused widespread floods within the

Danube and Elbe Rivers, as well as billions of euros in damages. The floods generated the largest summer discharge on record within the last 140 years. The high-frequency monitoring network in the German Bight available within the Coastal Observing System for Northern and Arctic Seas (COSYNA) captured the flood influence on the German Bight. Monitoring data from a FerryBox station in the Elbe Estuary (Cuxhaven) and from a FerryBox platform aboard the *M/V Funny Girl Ferry* (travelling

between Büsum and Helgoland) documented the salinity changes on the German Bight, which persisted for about 2 months after the peak discharge. The flood generated a large influx of nutrients, dissolved and particulate organic carbon on the coast. These conditions subsequently led to the onset of a chlorophyll bloom within the German Bight, observed by dissolved oxygen supersaturation, and higher than usual pH in surface coastal waters. The prolonged stratification also led to widespread bottom

water dissolved oxygen depletion, unusual for the south eastern German Bight in the summer.

1. **Rain saturated soils:** (May-June)

2. **Extreme Elbe discharge:** June

3. **Large freshwater influx, high nutrients, DOC & POC from Elbe River and onto the German Bight:** (June-July)

4. **2-month stratification; high primary production in surface; widespread DO depletion in bottom waters**

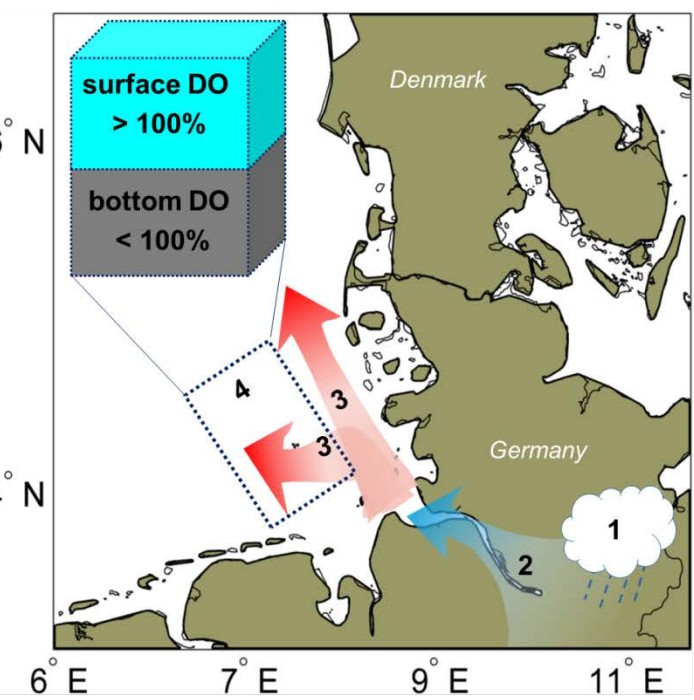



## 1 Introduction

General circulation models have predicted that the frequency of heavy rainfall events will increase over the next centuries with changes in climate (Karl et al., 1995; Meehl et al., 2007; Elsner et al., 2008; Bender et al., 2010), and particularly during summer months (Karl and Knight, 1998; Christensen and Christensen, 2004). Allan and Soden (2008) correlated climate models with satellite observations and concluded that extreme rainfall events and droughts will increase during warm months and these amplifications may be greater than current predictions. Depending on their magnitude, these abnormal physical conditions can cause phytoplankton blooms and disruptions in food webs (Paerl et al., 2006; Wetz and Paerl 2008). It is therefore essential that the impact of extreme rainfall and flood events on estuarine and adjacent coastal region biogeochemistry is better assessed (Scavia et al., 2002; Wetz and Yoskowitz, 2013).

More frequent occurrences of intense flood events and tropical cyclones are likely to generate large infrastructural damages as a result of flooding, high winds and higher storm surges (Wetz and Yoskowitz, 2013). For example, over recent decades, heavy floods (particularly in August, 2002 and June, 2013) in Europe have generated billions of euros in damage (Ionita et al., 2014; Merz et al., 2014). These stronger storm events can also lead to large-scale, prolonged stratification in estuaries and coastal regions (Hickel et al., 1993; Voynova and Sharp, 2012) as a result of large freshwater influxes. The stronger stratification and elevated nutrient and organic matter loading to estuarine and coastal systems, associated with these extreme climatic events, along with the projected increase in temperature already observed in some coastal ecosystems (Wiltshire and Manly, 2004; Luterbacher et al., 2016) could lead to the development of bottom water hypoxia (Statham, 2012; Voynova and Sharp, 2012; Wetz and Yoskowitz, 2013).

One of the most significant discharge events in Central and Western Europe took place in the summer of 2013 (Merz et al., 2014), and caused extensive flood damages on land due to large scale flooding in the southern and eastern parts of Germany, and the western regions of the Czech Republic (Ionita et al., 2014). The meteorological conditions preceding the flood have been extensively



documented. During May 2013, weather in and around central Europe was unusually cool and wet (Ionita et al., 2014), due to repeated upper-tropospheric Rossby wave-breaking, and the subsequent occurrence of a quasi-stationary upper-level cutoff low pressure system over Europe (Grams et al., 2014). Heavy precipitation (Global Precipitation Climatology Centre estimates) in Germany during the

5 last two weeks of May amounted to 100-200% of the expected climatological precipitation for the entire month. As a consequence, soils in most of the Elbe River catchment reached record levels of moisture by the beginning of June (Ionita et al., 2014; Grams et al., 2014). Additional heavy precipitation (75-100 mm) between 30 May and 3 June, caused by the passage of three cyclones, Dominik, Frederik, and Günther (Grams et al., 2014), fell over a number of countries including Germany, Austria, Switzerland,

the Czech Republic, and Poland (Merz et al., 2014). The inability of the already saturated soils to absorb the additional heavy precipitation, generated heavy flooding in the Elbe and Danube river basins (Grams et al., 2014; Ionita et al., 2014). A similar progression of events and increased soil moisture has been associated with two other major summer storms, in August, 1954 and August, 2002 (Merz et al., 2014).

This study focuses on the influence of the June 2013 flood on the biogeochemistry of the Elbe Estuary and the German Bight, as an example of the impact of extreme discharge events on the biogeochemistry of estuaries and adjacent coastal regions. The study compares the flood event to average conditions, by using a combination of existing historical datasets and high-frequency continuous measurements available from the Coastal Observing System for Northern and Arctic Seas

(COSYNA, Baschek et al., 2016). Whereas in 1954 and 2002, high-frequency autonomous monitoring of the German Bight was scarce, recent high-frequency monitoring platforms within COSYNA, along with other available historical datasets (from discrete sampling) have made it possible to capture the impact of a rare summer extreme discharge event. Considering that a future increase in storm events has been predicted by climate models with climate change, particularly in the summer, a supplemental

discharge analysis of the 140 year long Elbe River record has helped to assess the likelihood that such rare events may become more prevalent in the near future, and could significantly alter average coastal and estuarine biogeochemistry.



## 2 Methods

### 2.1 Study site

The relatively shallow (10-43 m) German Bight is situated in the southeast part of the North Sea, and its topography is dominated by the ancient Elbe River Valley (van Beusekom et al., 1999; Becker et

al., 1999). The Wadden Sea is a shallow coastal sea (<10 m), which borders the German Bight along the Dutch, German and Danish coasts (van Beusekom et al., 1999). The distribution of temperature and salinity in the bottom layers of the German Bight is strongly related to the topography, and follows the ancient Elbe River Valley (Becker et al., 1999). The German Bight is dominated by a counterclockwise residual circulation pattern, which carries a mixture of Atlantic water and continental runoff from the

Rhine and several other rivers into the German Bight from the west (Hickel et al., 1993; van Beusekom et al., 1999). The inflow of nutrients and contaminants from the Weser and the Elbe estuaries (van Beusekom et al., 1999), and the residual circulation favor the accumulation of contaminants in the German Bight (Hickel et al., 1993). While the central part of the North Sea is seasonally stratified due to both thermal and salinity differences, the shallow parts in the southeast German Bight and along the

Wadden Sea are well mixed due to strong tidal currents (Becker et al., 1999).

One of the largest rivers in Northern Europe, the approximately 1100 km long Elbe (Ionita et al. 2014), is the main source of freshwater to the southeast German Bight (Hickel et al., 1993) and stretches from Schmilka, Czech Republic to the German Bight. The riverine portion extends up to Geesthacht, Germany (Elbe km 580), where a weir marks the head of the tide, and separates the riverine from the

20 estuarine region (Petersen et al., 1999). The estuarine part, characterized by a salinity gradient, extends for about 125 km from Zollenspieker (599 km) to Cuxhaven (725 km) (Petersen et al., 1999; Petersen et al., 2000).

High nutrient loads, and damming of the river near Geesthacht, allow for nutrient assimilation by primary production in the nontidal riverine portion, generating high chlorophyll concentrations (> 60

25 µg L-1), dissolved oxygen supersaturation, and pH levels up to 9.5 in surface waters upstream of the dam weir (Petersen et al., 1999). Within the tidal freshwater region and the salinity gradient of the Elbe Estuary, however, as nutrients are regenerated from the large amounts of decomposing labile particulate carbon, oxygen levels can become severely undersaturated (and this defines the oxygen minimum zone





(OMZ)), particularly near Hamburg (Petersen et al., 1999; Amann et al., 2012); further downstream along the salinity gradient, oxygen levels increase, but typically remain undersaturated. Therefore a large degree of variation within the salinity gradient of the Elbe River is related to the production and processing of labile organic matter (Amann et al., 2012).

## 2.2 Data Sources

A number of stations, and moving or fixed monitoring platforms, shown in Fig. 1, and listed in Table 1, were used to understand the changes that occurred within the Elbe Estuary and the adjacent coastal regions in the southern part of the German Bight. Monthly maps of biogeochemical parameters of interest were generated using a combination of available measurements. For 2013, we focused on the months before (March), and after (July, August and September) the June flood event. The datasets used in the 2013 maps are listed in Table 1. The FerryBox and MARNET data was downloaded from the COSYNA data portal CODM (Breitbach et al., 2016).

### 2.2.1 River Discharge

A more than a century long Elbe River daily discharge record (1.11.1874 – 31.08.2015) is available from the Neu Darchau gauging station (Elbe km 536), located in the lower Elbe River catchment area, about 50 km upstream of the weir at Geesthacht (Fig. 1). The data are provided by the German Federal Waterways and Shipping Administration (WSV), and communicated by the German Federal Institute of Hydrology (BfG), and have been analyzed in terms of their daily, monthly and decadal distribution. In addition, a frequency analysis was performed on the 5-, 10-, 25-, and 50-year discharges to compare the number of occurrences within the last 15 years to the rest of the 140 year record.

### 2.2.2 Tidal Height at Cuxhaven, Germany

Sea level data (tidal height) were extracted from the GLOSS/CLIVAR database (http://www.gloss-sealevel.org/data/#.VxeHnUaFEak), from a station located near Cuxhaven, Germany (53.87 ° N, 8.72º W), which has been sampling between 1917 and 2015. The data are available in hourly, daily and monthly intervals. In this study we used the hourly dataset.





### 2.2.3 Cuxhaven FerryBox Station

A fixed FerryBox (Petersen et al. 2014) station has been operating at Cuxhaven (53.877º N, 8.705º W) since 2010, measuring temperature, salinity, dissolved oxygen (DO; Aanderaa optode), chlorophyll fluorescence (Chl; Turner Designs, Sunnyvale, CA), pH (Clark electrode) and turbidity

(Turner Designs, Sunnyvale, CA) approximately every 10 minutes.

All data from the Cuxhaven FerryBox station were resampled at an hourly interval; the 2012-2013 records had the most complete coverage of all parameters, and were therefore used for analysis in this study. Continuous dissolved oxygen optode data were corrected using six discrete samples taken between 2012 and 2014, and analyzed by Winkler titration. The Winkler titration data were on average

40.72 +/- 2.63 µM higher than the Aanderaa optodes, and thus the Cuxhaven DO optode 2012-2013 data were corrected by adding the average difference to the optode measurements.

Frequency analysis (Voynova et al., 2015) helped to identify a number of modes associated with tidal, daily and lower-frequency harmonics at this station. The frequency spectra for the FerryBox data were compared to sea level frequency spectra, so that biological signals could be identified. In addition,

isolating the signals associated with high tide and low tide allowed to better understand the biogeochemical changes at different locations in the Elbe Estuary.

### 2.2.4 Hamburg Port Authority (HPA) Elbe River Pile

The Cuxhaven FerryBox data were compared to data gathered at a pile operated by the HPA and Helmholtz-Zentrum Geesthacht (HZG), and deployed in the Elbe River (53.859º N, 8.944º W), about 15

20  km upstream of the Cuxhaven FerryBox station, during 2012 and 2013 (March - November). Every 10 minutes a variety of biogeochemical parameters, including temperature, salinity, pH, DO, chlorophyll fluorescence, and turbidity, were measured at the pile, and thus provided another reference station within the Elbe estuary. All HPA Elbe River data were resampled to an hourly interval.

The pH data at this station drifted considerably following the first 1000 hours of measurement.

The data were corrected by subtracting the moving average (window size of 1000 hours; Aguilera, 2008) for each year, and then adding the mean pH for those first 1000 hours of initial deployment. The pH signal corrections are available in the supplementary online materials (Fig. S1).



### 2.2.5 Funny Girl FerryBox

Throughout the summer months, from about May to September, the M/V Funny Girl ferry crosses the distance between Büsum and Helgoland in the German Bight (Fig. 1). Every day, the ferry departs from Büsum around 9:30, and then returns to Büsum around 18:15. A typical cruise lasts about 2 hours and 15 minutes, and there is a break of about 4 hours in Helgoland between the two crossings. On occasions, the ferry leaves Helgoland later, around 17:30, and arrives in Büsum around 20:00. A FerryBox installed aboard the ferry in 2008 measures a number of parameters including temperature, salinity, pH, chlorophyll fluorescence, dissolved oxygen, colored dissolved organic matter, and turbidity.

Not all parameters were available every year between 2008 and 2015. The longest available records (2008-2015) are for temperature, salinity and pH; the most complete records for every parameter are available for 2012, 2013 and 2014 seasons. These data were used to characterize the biogeochemical conditions in the German Bight in 2013, compared to non-flood years (2012 and 2014), and to quantify the influence of a summer flood event on the German Bight.

### 2.2.6 Deutsche Bucht (German Bight) Monitoring Station

Deutsche Bucht station (http://www.bsh.de/de/Meeresdaten/Beobachtungen/MARNET-Messnetz/Stationen/debu.jsp) is located east of Helgoland (53.167º N, 7.45º W), and measures salinity, water temperature, dissolved oxygen at a depth of 6 and 30 m, as well as meteorological parameters at a the surface and 14 m above the surface at hourly intervals. The meteorological data were not used in this study. The Deutsche Bucht station has been operated by the Federal Maritime and Hydrographic Agency of Germany (Bundesamt für Seeschiffahrt und Hydrographie (BSH)) since 1989, as part of the Marine Environmental Monitoring Network in the North and Baltic Seas (MARNET), and the data are also available on the COSYNA website. Historically a number of additional biogeochemical parameters, including phosphate, nitrate and nitrite, as well as silicate have also been measured at this station. Salinity, temperature and dissolved oxygen data for 2013 were used to understand the water column dynamics in the southern part of the German Bight.





### 2.2.7 Discrete Samples

Historically, BSH and the Biological Station Helgoland (BAH), of the Alfred Wegener Institute (AWI) have also collected discrete biogeochemical samples (surface and bottom) during routine monthly ship cruises throughout the German Bight over a number of years. Typical sampling station positions were shown in Fig. 1; however, not all stations were sampled every month. For this study, only the available discrete data for March, July, August and September were used, including salinity, temperature, phosphate, nitrate and nitrite, silicate, and ammonium measurements. In addition, surface and bottom dissolved oxygen samples available in August and September were compared to Deutsche Bucht data.

Finally, the BAH AWI data for the period between 2008 and 2015 (except 2013) were used to compile average conditions for nutrient and hydrographic parameters mentioned above, so as to have a reference for how different conditions were in the summer of 2013, after the extreme 2013 discharge event. Based on the average values for each month, monthly maps of nutrient distribution were generated based on Gaussian interpolation of the available data. In this study we focus on the distributions of salinity, nitrate, nitrite and silicate, to describe water mass characteristics, and parameters that are influenced by biological production.

### 3 Results

### 3.1 Discharge Analysis

The average discharge for the entire Elbe discharge record (Fig. 2) was $708 \pm 446$ m$^3$ s$^{-1}$. However there were distinct seasonal differences, and summer is typically the driest season (Fig. S2). Within the 140-year Elbe River discharge record (Fig. 2), the June 2013 outflow was the highest of all summer daily discharges, and overall the second highest discharge on record, with two daily flows of the same magnitude (4060 m$^3$ s$^{-1}$) on June 11 and 12, 2013. The highest overall daily discharge (4400 m$^3$ s$^{-1}$) was recorded on March 25, 1888. The June 2013 discharge was so large, that the average discharge for the entire month of June, 2013 was also significantly elevated (Fig. S2a), compared to the June discharge over the rest of the 140-year old record (monthly means).



In recent decades (1966 – 2015, Fig. S2b), there was a possible change in the monthly precipitation distribution, so that most of the elevated discharges in the spring period were distributed over January, February, March and April, instead of concentrated during March and April. Also, the magnitude of the average monthly spring discharge during the last 4 decades was smaller than the March-April decadal averages prior to 1945 (Fig. S2b). This suggests that there is a redistribution of average monthly discharge patterns within the last 4 decades.

The recurrence period of the highest June daily discharge is every 70 years, when considering the entire 140-year discharge record. However, between 1915 and 2015, the June 2013 event (in terms of daily discharge) was the largest flood; therefore it could also be considered as a 100-year storm. Depending on the return period (5, 10, 25, or 50 years, Table 2), 20 to 60% of the large to extreme daily discharges occurred in the last 15 years (2001-2015), resulting in a significant increase in the frequency of large to extreme discharges.

## 3.2 Flood Influence in the Elbe Estuary

In order to understand the influence of the large June 2013 discharge on the Elbe Estuary, the hourly Cuxhaven FerryBox station data were plotted alongside the HPA Elbe Pile data for 2012-2013. Although spring discharge during both years was about the same, the 2012 summer discharge was considerably lower ($< 1000$ m$^3$ s$^{-1}$, Fig. 3), which is reflected in the higher average salinities observed at both stations in 2012. The 2012 data could be used as a reference for the biogeochemical patterns during a dry season, while 2013 represents an extreme discharge summer.

Despite the stations' relative proximity (~ 15 km distance), Cuxhaven FerryBox and HPA Pile were located in two distinct regions in the estuary, and the biogeochemical parameters in Fig. 3 also varied considerably over a tidal cycle, over the summer season, and between 2012 and 2013. A frequency analysis of all available data at Cuxhaven (Fig. 4) allowed us to identify the main frequencies associated with each variable in Fig. 3. As a reference, the sea level power spectral density at the FDH station near Cuxhaven is shown next to each parameter (Fig. 4). All parameters (temperature, salinity, DO, Chl, turbidity, pH, sea level) have a pronounced peak associated with the 12.5 hour tidal period (most likely $M_2$ and $S_2$ lunar and solar semi-diurnal constituents), as well as with the residual shallow



tidal 8, 6 and 4 hour periods (Voynova et al., 2015). In addition, a smaller frequency peak is resolved at the 24-25 hour period, most likely associated with the day-night cycles and the $O_1$ and $K_1$ lunar diurnal tidal constituents. The 24-hour peak is slightly more pronounced in the DO and temperature plots (Fig. 4), which suggests that these parameters are affected by the day-night cycles in temperature and primary

production. Finally, the 60-hour window size did not allow for resolving lower frequencies like spring-neap variability or storms. However, these low-frequency modes likely influence the biogeochemistry in the Elbe Estuary, including the seasonal changes in water temperature (Fig. 3).

In order to better visualize the flood influence, and considering the large tidal ranges (~ 3 m at Cuxhaven) in the Elbe Estuary, the position of salinity minima and maxima at each station were

identified, based on methods described in Voynova et al. (2015). The values of several parameters (salinity, temperature, dissolved oxygen, pH, and chlorophyll fluorescence) at the identified positions were extracted to represent the flood and ebb water mass end members at Cuxhaven and HPA Pile (Fig. 5). About 4-5 days after the peak discharge (June 12-13) in 2013 caused a pronounced salinity decrease at Cuxhaven, the ebb tide salinity dropped below 3 for about 8 days, while the flood tide salinity

dropped to about 10 on June 18. The elevated discharge shifted the entire salinity gradient seaward, so that around river km 710 (HPA Pile), salinity dropped to below 2 for at least 9 days (Fig. 5).  It is important to note that the ebb salinity at Cuxhaven was very similar to the flood salinity at the HPA Pile, which suggests that water was usually transported between the two monitoring stations over a tidal cycle.

The flood influence in the Elbe Estuary was prolonged, as suggested by the gently sloping falling limb of the storm hydrograph, and the depressed salinity at both stations between the beginning of June and the end of July, 2013 (Fig. 5). Between June and July, 2013, the tidal salinity range at Cuxhaven increased and was close to double the typical range in 2012, while the salinity range at HPA Pile decreased to about half the typical range.

Several biogeochemical parameters were also influenced by the flood. Dissolved oxygen decreased, when salinity dropped (down to 65% saturation at HPA Pile), suggesting the delivery of oxygen-depleted low salinity water from riverine tidal regions (Amann et al., 2012). Dissolved oxygen bounced back to pre-storm levels, and then increased to close to saturation, likely associated with an





increase in local production after the storm. During the flood, oxygen fluctuations were diminished. Changes in pH tracked dissolved oxygen before the storm, but as salinity started to decrease, pH also decreased to 7.5 (HPA Pile, and ebb tide at Cuxhaven), which suggests that the change in water mass affected both DO and pH. After the storm, pH was still depressed (< 8, Fig. 5), but also tracked DO.

By separating the flood from ebb data, we can better visualize the changes in DO related to each water mass end member. In both 2012 and 2013, DO was typically highest during flood tide at Cuxhaven, and it sometimes supersaturated in surface waters. This indicates that while upstream estuarine regions are generally DO-depleted (Amann et al., 2012), in the coastal regions adjacent to the Elbe Estuary, DO is supersaturated due to high primary production rates. While DO supersaturation also

took place in the spring/early summer, 2012, coincident with elevated pH (> 8), in 2013, the highest DO during 2013 was measured in July and August. This suggests a possible increase in primary production after the flood event.

In 2013, high chlorophyll concentrations (Fig. 5) during the flood tide at Cuxhaven coincided with the highest pH values, which suggests that there was a large bloom in the coastal waters near

Cuxhaven at the end of May, beginning of June, perhaps stimulated by the elevated precipitation and discharge during May 2013 (Merz et al., 2014). After the onset of the June flood, chlorophyll concentrations abruptly decreased and were lowest around the time of lowest salinity (5 days after the peak discharge at Neu Darchau), DO and pH, suggesting that the bloom was flushed out with the surge of freshwater and a large amount of potentially labile organic matter was transported to the German

Bight.

**3.3 Load Calculations**

The residence time of the Elbe estuary significantly decreases with increase in river discharge, based on analysis done by Bergemann et al., (1996). For a discharge of 250 m$^3$ s$^{-1}$ it takes 84 days to flush out the estuary; for 700 m$^3$ s$^{-1}$ the flushing time decreases to 30 days, and for 1200 m$^3$ s$^{-1}$ down to

18 days. We can obtain a linear correlation between the log of the discharge and the log of residence time (Eq. (1), R$^2$ = 1.00):

$$Log(Residence\ Time) = -0.98 * Log(Discharge) + 4.28 \qquad (1)$$



Following this relationship, we can estimate that for an extreme discharge like June 2013 (2 consecutive days of 4060 m$^3$ s$^{-1}$), the flushing time dropped down to 5 days. This number coincides with the delay in salinity decrease, observed 4-5 days after the peak June discharge (Fig. 5). Therefore, rather than being processed within the estuary, during the flood, particulate and dissolved organic carbon and

nutrients from the riverine region were instead delivered relatively quickly to the adjacent coastal regions in the German Bight.

BSH calculated that between 12 June and 8 July, 2013, nutrient loading near Hamburg was significantly elevated (Table 3). Nutrient loads varied with time (Weigelt-Krenz et al., 2014): while the peak nitrate and silicate loading coincided with the peak discharge (12-16 June), the peak ammonium

and phosphate loading occurred between June 24 and 28, about 12 days later. Therefore the influx of ammonium and phosphate into the higher salinity regions of the estuary and into the German Bight was delayed compared to nitrate and phosphate.

Using historical total organic carbon (TOC), total suspended solids (TSS) and salinity data (1992-2013) measured biweekly and monthly (1992 – 2013) next to the Cuxhaven FerryBox station by

River Basin Community Elbe (FGG Elbe, http://www.fgg-elbe.de/start-en.html), a strong negative correlation ($R^2 = 0.53$) was generated between TOC and salinity (Fig. S3): TOC = -24*Salinity + 886. There was also a strong positive linear correlation ($R^2 = 0.50$) between total suspended sediments and total organic carbon: TOC = 3*TSS + 311. This suggests that floods generate large decreases in salinity (< 15) at Cuxhaven transport large amounts of sediments with increased amount of carbon (> 400 µM)

from the Elbe River and onto the adjacent coastal waters in the German Bight.

To summarize, the extreme discharge event caused a shift in the entire salinity gradient of the Elbe Estuary, and salinity was overall depressed compared to typical levels for more than a month. The heavy rains in the second part of May (Ionita et al., 2014), and subsequent elevated discharge, had generated a nutrient influx, and a large bloom near Cuxhaven. When this bloom was flushed out during

the extreme June discharge, it was a large source of labile organic material, and additional nutrient loading onto the German Bight.



### 3.4 2013 Flood Influence on the German Bight

To examine the influence of the 2013 June flood on the German Bight, we used several data sources listed in Table 4. The most extensive records of the changes on the coast were available from the *M/V Funny Girl* FerryBox (Fig. 6), which measures temperature, salinity, chlorophyll fluorescence,

DO, colored dissolved organic matter (CDOM) fluorescence, and pH (2 Clark electrodes). The data allowed to contrast an anomalous year (2013) to drier summer conditions (2012 and 2014). The region between Büsum and Helgoland was very dynamic in the summer, with a typical salinity range of about 5-6, and a temperature range of up to 5 degrees between the two ports. In addition, although DO varied seasonally, it was often supersaturated along the ferry transect, suggesting that the region between

Büsum and Helgoland is productive between April and October.

Even though CDOM fluorescence was not calibrated against discrete DOC samples, the CDOM range was similar in 2012 and 2014 (Fig. 6). In 2012 and 2014, and before the flood in 2013, CDOM varied linearly with salinity along the ferry transect (Fig. 7), indicating dilution of continental allochtonous sources of dissolved organic carbon, without a significant source or sink of dissolved

organic carbon. The similarity of the slopes for 2012, 2013 and 2014 also indicated that interannual variation of dissolved organic matter was a function of dilution of freshwater sources of organic carbon.

The highest pH values typically occurred in the beginning of the summer season, or the end of the spring season. In 2012, and 2014, this indicates perhaps the end of the spring bloom. In 2012 and 2014, pH had a pronounced seasonal drift reflected in the records of both pH electrodes, so that the

lowest pH typically occurred in the fall (Fig. 6). This suggests that pH differs from DO trend in this region, and that the two parameters should be used with caution for examining primary production and respiration trends.

The June 2013 flood caused significant changes in all parameters in Fig. 6. While temperature increased slightly, salinity in the middle of June, 2013 decreased dramatically to below 15 near the

Wadden Sea and remained depressed through July and August; at the same time, salinity range increased to about twice the range during 2012 and before and after the flood in 2013. The large outflow from the Elbe estuary was also tracked using a number of biogeochemical parameters (Fig. 7). At first the lower salinity water mass that reached the German Bight was characterized by high chlorophyll



concentrations, associated with seaward flushing of the coastal bloom observed near Cuxhaven. Then, decreasing salinity in the German Bight, probably tracked water outflow from the Elbe Estuary, which was undersaturated and loaded with large amounts of CDOM, up to double the typical concentrations observed in 2012 and 2014 (Figs. 6 and 7). This indicates that a large pulse of dissolved organic carbon was quickly delivered on the coast. Also, the slight deviation from linearity in CDOM with respect to salinity after the flood (Fig. 7) indicates a potential autochtonous source of dissolved organic matter along the ferry transect.

The storm discharge also caused the lowest salinity on record (Fig. 8) in all available *M/V Funny Girl* data (2007-2014) particularly near the coastal Wadden Sea (North Frisian coast). At the end of June and beginning of July, the entire ferry transect was fresher than usual, and this suggests a delayed effect from the plume on the coast compared to the salinity changes observed in Cuxhaven, due to transport of estuarine water offshore. The east Wadden Sea (up to about the middle of the ferry transect (Table 1), between Büsum and Helgoland) was fresher than usual (salinity < 25) throughout the whole 2013 season, which suggests that heavy precipitation in May prior to the storm (Ionita et al., 2014) also affected salinity on the coast (Fig. 8), and probably helped to generate the chlorophyll bloom, observed in May, 2013 and associated with DO supersaturation and elevated pH (Fig. 6). The cooler spring weather was reflected in the unusually cold surface water temperatures during May and into June (5-15º C), compared to the rest of the summer records, especially near Helgoland.

Based on the 7-year record (Fig. 8) from *M/V Funny Girl*, pH was usually high in the beginning of summer, probably due to high biological production during the spring bloom. Later in the year, pH typically decreased, and the lowest pH values occurred at the end of the summer, beginning of fall. During summer, 2013 however, high pH (> 8) persisted throughout spring and into early June. Then, after a brief period of low pH, which coincided with the lowest salinity water, the July pH values increased to about spring bloom levels. Compared to the typical pattern observed during all other years, the unusually high pH late in the summer (July-August) was most likely associated with a coastal bloom which formed after the flood, in response to the nutrient influx and potential stratification of the water column. Even though the seasonal patterns of the two parameters differed in this region, DO supersaturation in July also supports this suggestion. All of these factors suggest that the June 2013



storm event had a substantial and rare effect on the German Bight between Helgoland and Büsum, which had not been observed within the last 7 years.

In combination with discrete and autonomous sampling from BSH and Helgoland (see Table 4), the *M/V Funny Girl* and Cuxhaven FerryBox data were used to create maps for March, July, and August for surface salinity, nitrate, nitrite, and silicate (Figs. 10, 11, 12). The data for these months was the most complete between all data sources, allowing for more detailed surface maps to be generated. The 2013 maps were compared to average distributions of all parameters, generated from the average monthly samples collected by AWI (2008-2015, excluding 2013). The March 2013 parameter distributions were similar to average conditions in patterns and magnitude, especially for salinity and nitrate and nitrite. In July, following the June 2013 large discharge event, there was a large plume of low salinity (< 28), high nitrate (3-43 µM) water along the coastal regions near the western Wadden Sea, and extending north along the coast and west and slightly south of Helgoland. The plume thus spread well over the whole Helgoland Bight in July, about a month after the large discharge event. The plume also carried higher concentrations of ammonium (not shown) and silicate onto the coastal shelf regions, although their patterns differed slightly from the salinity distributions. The July 2013 maps were quite different from average high salinity (> 25) and low nitrate (0.01-1 µM) patterns typically found in July (Fig. 11).

To better understand the influence of the freshwater plume on water column stratification and dissolved oxygen distribution, we used temperature, salinity and DO data from the Deutsche Bucht MARNET station, located east of Helgoland (Fig. 13). This station was affected by the low salinity, high $NO_3 + NO_2$ plume in Fig. 11. Although some data were missing for July 2013, surface water temperature and salinity at the end of July and during August differed from bottom distributions, suggesting the establishment of persistent water column stratification. Even though the vertical temperature gradient decreased after the middle of August, the presence of a low salinity surface water probably helped to maintain stratification up to September. Surface dissolved oxygen supersaturation suggests that in July and August, there was enhanced production within the surface mixed layer, and respiration of locally produced organic matter in the isolated bottom waters, which reduced dissolved oxygen at depth (Fig. 13). Even though supersaturation also occurred in the spring (April-May),





dissolved oxygen in bottom waters was only undersaturated in the latter part of the summer, after the water column remained stratified for about two months. At the end of September, and beginning of October, after stratification broke (there was no vertical gradient in temperature and salinity), DO in surface and bottom waters equilibrated to about saturation levels.

In 2013, the most complete summer discrete dissolved oxygen records within the German Bight were available in August and September from surface and bottom samples, measured by BSH and AWI cruises (Fig. 14). Most of the surface samples in August (93 %) were supersaturated, indicating high primary production in surface waters throughout the southeast German Bight; in September, only 6% of the surface samples were supersaturated. Bottom water DO undersaturation suggests that prolonged

water column stratification established within the German Bight. In August, 2013, 71% of the bottom oxygen measurements were undersaturated, and 42% of the stations measured DO < 85% saturation. In September, 91% of the bottom samples were undersaturated and 40% experienced DO saturation of 85% or less. The maps in Fig. 14 suggest that in the German Bight, especially within the Elbe River valley, and east of Helgoland near the Deutsche Bucht station, bottom dissolved oxygen was

undersaturated in both August and September. The discrete sample data combined with the continuous observations made by the fixed Deutsche Bucht station suggest that the observed stratification and dissolved oxygen depletion in bottom waters was widespread within the southeast German Bight, and persisted at least 2-3 months after the extreme June discharge.

        To summarize, the June 2013 discharge generated a large plume of low salinity waters from the

Elbe Estuary and over most of the southeast German Bight, and carried large amounts of nutrients and dissolved and particulate organic carbon onto the coastal regions. The storm outflow affected primarily the coastal regions near the western Wadden Sea, and spread north of Büsum along the coast, but also west and south of Helgoland. The flood plume was present in July and August, and caused persistent stratification on the coast. The influx of nutrients and the establishment of atypical prolonged water

column stratification increased primary production in the surface mixed layer, as evidenced by oxygen supersaturation (> 110%) in surface waters, and also contributed to widespread oxygen depletion in the isolated bottom layers.





## 4 Discussion

The June 2013 discharge event was the second largest in the 140 year discharge record of the Elbe River at Neu Darchau, and had a significant influence on the biogeochemistry of the Elbe Estuary. The residence time in the estuary probably decreased to 4-5 days, and even though salinity changes at

the mouth of the Elbe Estuary were smaller compared to upstream regions, there was a notable constriction of the salinity gradient, and larger salinity fluctuations over a tidal cycle at Cuxhaven station during and after the storm. Based on available observations it can be deduced that a large bloom in the coastal waters near Cuxhaven was flushed out by the storm outflow onto the shelf near the western Wadden Sea.

The doubling of CDOM fluorescence detected by the ferry, suggests that the low salinity water plume carried a large load of continental-based dissolved organic matter which may have doubled the typical concentrations observed on the coast. The average DOC concentrations in the freshwater tidal river of the Elbe Estuary are relatively high, 500-600 µmol L$^{-1}$ (Amann et al., 2012), and with the low processing time, and decreased residence time, a large portion of that DOC pool was transported on the

coast during the flood. It is estimated that 25-33% of the dissolved and particulate continental-based organic carbon sources are labile, and contribute to rapid remineralization on the continental shelves and shelf seas (Smith and Hollibaugh, 1993). Typically, about one third of the organic matter loads into the Ems Estuary and the Western Dutch Wadden Sea is derived from freshwater sources (van Buesekom and de Jonge, 1998). With increased carbon loading from the Elbe River in summer 2013,

river-borne sources of labile organic carbon from freshwater and coastal phytoplankton blooms probably increased significantly, and altered the carbon cycle in the west Wadden Sea and adjacent coastal regions.

In addition, the slight deviation from linearity in the post–flood salinity vs. CDOM plot suggests that there may have been a source of autochthonous dissolved organic carbon on the coast, likely due to

remineralization of the increased allochtonous particulate organic carbon load after the flood. In the last 25 years, with decreasing pollution, the amount of particulate organic carbon in the Elbe Estuary has increased from 10 to 30 % of the total organic carbon pool, and about 50% of the POC is efficiently remineralized in the oxygen minimum zone of the Elbe Estuary before reaching the turbidity maximum;



the rest of the POC pool is remineralized in the turbidity maximum (Amann et al., 2012), located upstream of the HPA Pile in Fig. 1. A large flood event, like the June 2013 extreme discharge substantially decrease the residence time of the estuary, and shorten the time for remineralization of POC, thus delivering a substantial amount of continental-based organic carbon to the shelf where it can

contribute to respiration (Cai et al., 2011). The observed decrease in oxygen after the flood (Fig. 8) for example may have also been the result of increased respiration of allochtonous labile organic carbon. An extreme event like the 2013 June discharge can therefore alter substantially the carbon sinks and sources in adjacent coastal and shelf seas, and should be considered when calculating carbon budgets in coastal systems.

In addition to carbon loading, the flood event delivered large amounts of nitrogen from the Elbe estuary to the southeast German Bight. Despite the significant decrease of ammonium loads since 1989 (Petersen et al., 1999), nitrogen loading in the Elbe in the form of nitrate is still high (> 150 µM about 15 km from the mouth of the estuary), and represents a significant nutrient source to the German Bight (Hickel et al., 1993). The June 2013 discharge generated nutrient loads in the estuary that were 2-50

15    times higher than average nutrient loads, measured for the month of June between 1996 and 2005 (Table 3; Weigelt-Krenz et al., 2014). The large scale nutrient loading spread into a large portion of the German Bight, extending north along most of the western Wadden Sea, as well as south and west of Helgoland, in regions that are typically nitrogen depleted during the summer (Fig. 11-12). The plume was observed up to 1-2 months after the flood, in both surface salinity and nutrient distributions. The

sudden nitrogen influx stimulated growth of primary producers in the surface waters, which is supported by the dissolved oxygen supersaturation measured by *M/V Funny Girl* after the flood event. By August, nitrate and nitrite concentrations were much lower, suggesting efficient uptake of nitrogen in the German Bight.

        Typically, in summer, when nutrient influx from rivers is reduced, remineralization of organic

matter plays an important role in sustaining high primary production in the German Bight. The annual turnover rate in the North Sea and the Wadden Sea is high (van Beusekom et al., 1999; Brockmann et al., 1999, Reimer et al., 1999). Therefore, a sudden influx of nitrogen-rich water on the coast is likely to stimulate the already efficient high rates of primary production and remineralization (van Beusekom et



al., 1999). The faster rates of phosphate remineralization (Hickel et al., 1993) probably helped to sustain increased primary production, despite the low phosphate influx after the flood event (Fig. 12). This was observed in 2013, as more than 90% of the surface measurements of dissolved oxygen in August were supersaturated, even up to 2 months after the extreme discharge.

The June 2013 flood had two important effects on the coastal carbon cycle. On one hand the freshwater plume delivered allochtonous organic carbon to the coast that is otherwise typically processed within or near the Elbe estuary. This organic carbon influx stimulated remineralization on the coast. On the other hand, the nutrient influx in the German Bight stimulated phytoplankton growth, and the increased production of autochtonous organic carbon. Both of these processes probably affected the

coastal carbon cycle up to 2-3 months after the flood event. A further implication may be a longer term effect from the flood event on the nutrient budgets and eutrophication state of the German Bight. Hickel et al. (1993) suggested that eutrophication (and changes in nutrient loads) in the inner regions of the German Bight may have a delayed effect of up to several years on the outer German Bight through the transport and subsequent recycling of organic plankton and detritus. Therefore the June 2013 discharge

may have had an even more prolonged and widespread effect on the ecosystem of the German Bight than has been emphasized in this study. To further investigate this, it is necessary to do a long term study on the carbon dynamics at existing stations, and include more detailed analysis of the carbon sinks and sources in the southeast German Bight and adjacent regions.

       Near the Deutsche Bucht station, the water column stratified, and stratification and the presence

of a bloom in the surface waters resulted in the undersaturation of dissolved oxygen in the bottom waters up to 2 months after the discharge. From a number of additional discrete samples (Fig. 14), stratification and dissolved oxygen depletion seemed to have been widespread throughout the regions affected by the plume. As a reference, Topcu and Brockmann (2015) found that the mean bottom water dissolved oxygen in the German Bight has a saturation rate between 83.9 and 99.6%. Most of the

bottom water dissolved oxygen (% saturation) was much lower in August and September, 2013. This suggests that bottom water oxygen depletion and hypoxia in the southeast German Bight may be another detrimental effect from an extreme discharge event. One of the reasons for the persistence of water column stratification was probably due to the overall stable conditions on the coast. Callies et al.




(2016) used the results from a principal component analysis of the daily model output of the residual circulation in the German Bight and determined that during most of summer 2013 conditions were stable, with overall low wind conditions. Hickel et al. (1993) similarly observed that calm wind conditions after a flood event (summer 1981) allow the development of stratification and favorable light

conditions for phytoplankton growth, whereas strong winds lead to vertical mixing, poor light conditions and no bloom after a flood event (winter-spring 1987-88).

The large scale influence and potential long-term effects from extreme discharges on estuarine and coastal systems may become more frequent with changes in climate (Statham, 2012; Voynova and Sharp, 2012), and the June 2013 discharge and its influence on the German Bight serves as an excellent

example. Although average summer precipitation is predicted to decrease within the next 100 years, extreme precipitation events are expected to increase (Christensen and Christensen, 2004), and this is what has been observed in the discharge patterns of major rivers like the Elbe. Up to 20-60 % of the very large and extreme discharge events have taken place in the last 15 years, and two of the largest discharges took place during summer, in August, 2002 and June, 2013. Water temperature increases in

two of the major northern European river basins, the Elbe and the Danube Rivers have already been observed as a response to air temperature increase driven by climate change (Markovic et al., 2013), and summer air temperatures in recent years have been the highest on record over the past 2000 years (Luterbacher et al., 2016). Therefore, as we are already seeing the changes that have been predicted with climate change models (Karl et al., 1995; Allen and Soden, 2008; Bender et al., 2010), it is

important to better prepare for how to study and manage coastal systems affected by these extreme events. Whereas large spring flood events may be predicted based on snowpack and snowmelt characteristics months before the discharge, summer discharges generated by large precipitation events are more difficult to predict in advance (Ionita et al., 2014). It is useful to have monitoring networks like COSYNA in place, which can be further expanded with biogeochemical parameters, like bottom

dissolved oxygen sensors to help track the state of the ecosystem before and after an extreme event. In addition, further studies of carbon and nitrogen species could help determine the immediate and more long-term effect of these extreme events on the carbon and nitrogen cycles in coastal ecosystems.



## 5 Conclusions

The influence of the 2013 extreme June Elbe River discharge on the Elbe Estuary and the adjacent German Bight was captured using discrete samples and real-time COSYNA continuous monitoring platforms. This flood event could serve as a well-documented example of how extreme
discharges can alter the biogeochemistry of estuarine and adjacent coastal areas. The flood delivered increased loads of particulate and dissolved organic carbon, and nutrients on the coast. The increased loading of labile organic carbon most likely altered the coastal carbon cycle, as observed by the initial decrease in dissolved oxygen and pH shortly after the flood event in July, suggesting increased respiration of organic matter. Also, up to 2 months after the flood, water column stratification and
enhanced primary production, as evidenced by high pH and prolonged dissolved oxygen supersaturation in surface waters throughout the southeast German Bight, caused a more long-term and widespread effect on the coast. Finally, the atypical depletion of dissolved oxygen in the stratified bottom waters could be another potentially detrimental effect on coastal ecosystems. Bottom water oxygen depletion can be enhanced in the summer, when temperature is high and reaction rates are fast. Since large and
extreme floods have increased in frequency in recent decades, and 20-60 % of them (depending on discharge magnitude) have occurred in the last 15 years, the biogeochemical changes described in this study may become more prevalent in the future, and particularly during summer months. This effect of climate change has already been observed in a number of watersheds. Establishing continuous monitoring platforms then becomes essential for quantifying the influence of these events on coastal
and estuarine biogeochemistry.

## 6 Author Contribution

Y.G. Voynova, H. Brix and W. Petersen contributed to the study design and conception and drafting of the initial manuscript, and all authors listed contributed to analysis and interpretation of the
25 data, editing and critical revision of the final draft of the manuscript. W. Petersen, H. Brix, S. Weigelt-Krenz, and M. Scharfe contributed to data collection for COSYNA, and for the discrete sample datasets.




Y.G. Voynova performed the calculations, created the graphic material, managed the drafting of the manuscript and coordinated the other author's contributions.

## 7 Acknowledgements

We would like to thank the FerryBox team at HZG for data collection and maintenance of the FerryBox systems, as well as all people responsible for the COSYNA data collection. Also, we would like to thank the Nutrients team at BSH, who collected and analysed the discrete samples for nutrients, dissolved oxygen, salinity and temperature, as well as the Biosciences and Shelf Sea System Ecology teams at AWI in Helgoland for collecting and analysing the discrete samples along the Helgoland cruises. We would like to acknowledge the German Federal Waterways and Shipping Administration (WSV), and the German Federal Institute of Hydrology (BfG) for providing the discharge data from the Elbe River, and the River Basin Community Elbe (FGG Elbe, http://www.fgg-elbe.de/start-en.html) for providing the data for TOC, TSS and chlorophyll near the mouth of the Elbe River. Finally we would like to acknowledge the GLOSS/CLIVAR database (http://www.gloss-sealevel.org/data/#.VxeHnUaFEak), from where we obtained the sea level height for Cuxhaven tidal station. This work has been supported through the Coastal Observing System for Northern and Arctic Seas (COSYNA).





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





Table 1. Data sources: sampling dates, position, depth and parameters measured at different stations or moving platforms in the Elbe Estuary and the German Bight. BAH stands for the Biological Station Helgoland, at the Alfred Wegener Institute (AWI); BSH stands for Federal Maritime and Hydrographic Agency of Germany (Bundesamt für Seeschifffahrt und Hydrographie); HPA stands for Hamburg Port Authority; HZG stands for Helmholz-Zentrum Geesthacht.* Station Elbe 9 is located at about the same position as Elbe 3.

| Station Platform | Organization | Latitude | Longitude | Station Depth | Measurement Depth | Time Frame | Frequency | Salinity Temp | Nutrients | DO | Chl | Bottom data |
|---|---|---|---|---|---|---|---|---|---|---|---|---|
| Elbe 1 | BAH | 54.15 | 7.89 | 51 | 1 | 2008-2015 | monthly | yes | yes | no | no | yes |
| Elbe 2 | BAH | 54.10 | 7.99 | 28 | 1 | 2008-2015 | monthly | yes | yes | no | no | yes |
| Elbe 9* | BAH | 54.05 | 7.99 | 28 | 1 | 2008-2015 | monthly | yes | yes | no | no | yes |
| Elbe 3 | BAH | 54.05 | 8.08 | 20 | 1 | 2008-2015 | monthly | yes | yes | no | no | yes |
| Elbe 4 | BAH | 54.01 | 8.24 | 20 | 1 | 2008-2015 | monthly | yes | yes | no | no | yes |
| Elbe 5 | BAH | 53.99 | 8.31 | 18 | 1 | 2008-2015 | monthly | yes | yes | no | no | yes |
| Elbe 6 | BAH | 53.98 | 8.41 | 18 | 1 | 2008-2015 | monthly | yes | yes | no | no | yes |
| Elbe 7 | BAH | 53.95 | 8.50 | 15 | 1 | 2008-2015 | monthly | yes | yes | no | no | yes |
| Elbe 8 | BAH | 53.90 | 8.67 | 18 | 1 | 2008-2015 | monthly | yes | yes | no | no | yes |
| Eider 1 | BAH | 54.18 | 7.95 | 10 | 1 | 2008-2015 | monthly | yes | yes | no | no | yes |
| Eider 2 | BAH | 54.18 | 8.04 | 29 | 1 | 2008-2015 | monthly | yes | yes | no | no | yes |
| Eider 3 | BAH | 54.21 | 8.15 | 20 | 1 | 2008-2015 | monthly | yes | yes | no | no | yes |
| Eider 4 | BAH | 54.23 | 8.31 | 12 | 1 | 2008-2015 | monthly | yes | yes | no | no | yes |
| Eider 5 | BAH | 54.23 | 8.40 | 10 | 1 | 2008-2015 | monthly | yes | yes | no | no | yes |
| Eider 6 | BAH | 54.22 | 8.48 | 7 | 1 | 2008-2015 | monthly | yes | yes | no | no | yes |
| Eider 7 | BAH | 54.16 | 8.37 | 11 | 1 | 2013-2015 | monthly | yes | yes | no | no | yes |
| Eider 8 | BAH | 54.05 | 8.43 | 11 | 1 | 2013-2015 | monthly | yes | yes | no | no | yes |
| P8 1 | BAH | 54.15 | 7.89 | 53 | 1 | 2008-2015 | monthly | yes | yes | yes | no | yes |
| P8 2 | BAH | 54.18 | 7.79 | 41 | 1 | 2008-2015 | monthly | yes | yes | yes | no | yes |
| P8 3 | BAH | 54.16 | 7.67 | 34 | 1 | 2008-2015 | monthly | yes | yes | yes | no | yes |
| P8 4 | BAH | 54.15 | 7.57 | 35 | 1 | 2008-2015 | monthly | yes | yes | yes | no | yes |
| P8 5 | BAH | 54.25 | 7.38 | 37 | 1 | 2008-2015 | monthly | yes | yes | yes | no | yes |
| P8 6 | BAH | 54.27 | 7.19 | 36 | 1 | 2008-2015 | monthly | yes | yes | yes | no | yes |
| FerryBox Funny Girl | HZG COSYNA | 54.17-54.13 | 7.91-8.82 | varied | 1 | 2008-2015 | May-Sept (moving) | yes | no | yes | yes | no |
| FerryBox Cuxhaven | HZG COSYNA | 53.88 | 8.71 | | 1 | 2010-2015 | 10 minutes | yes | yes | yes | yes | no |
| HPA Pile | HPA, HZG | 53.86 | 8.94 | | 1 | 2012-2013 | 10 minutes | yes | no | yes | yes | yes |
| Deutsche Bucht | BSH | 54.17 | 7.45 | 30 | 6 | 2013 | hourly | yes | no | yes | no | yes |
| MEDEM | BSH | 53.88 | 8.72 | 16 | 1 | 2013 | 4 times/yr | yes | yes | no | no | yes |
| ELBE1 | BSH | 54.00 | 8.11 | 24 | 6 | 2013 | 4 times/yr | yes | yes | no | no | yes |
| WESER | BSH | 53.85 | 8.00 | 17 | 5 | 2013 | 4 times/yr | yes | yes | no | no | yes |
| STG 16 | BSH | 53.94 | 7.40 | 25 | 5 | 2013 | 4 times/yr | yes | yes | no | no | yes |
| HLOCH | BSH | 54.08 | 7.83 | 43 | 5 | 2013 | 4 times/yr | yes | yes | no | no | yes |
| HELGO | BSH | 54.25 | 8.10 | 18 | 5 | 2013 | 4 times/yr | yes | yes | no | no | yes |
| EIDER | BSH | 54.23 | 8.38 | 14 | 5 | 2013 | 4 times/yr | yes | yes | no | no | yes |
| KS11 | BSH | 54.07 | 8.13 | 20 | 5 | 2013 | 4 times/yr | yes | yes | no | no | yes |
| UE28 | BSH | 54.50 | 8.20 | 13 | 5 | 2013 | 4 times/yr | yes | yes | no | no | yes |
| AMRU2 | BSH | 54.67 | 7.83 | 14 | 5 | 2013 | 4 times/yr | yes | yes | no | no | yes |
| URST2 | BSH | 54.67 | 7.50 | 23 | 6 | 2013 | 4 times/yr | yes | yes | no | no | yes |
| URST1 | BSH | 54.42 | 7.58 | 28 | 6 | 2013 | 4 times/yr | yes | yes | no | no | yes |
| UFSDB | BSH | 54.18 | 7.43 | 39 | 5 | 2013 | 4 times/yr | yes | yes | no | no | yes |
| HPAE3 | BSH | 54.05 | 7.97 | 31 | 5 | 2013 | 4 times/yr | yes | yes | no | no | yes |





Table 2. Number of daily discharges ($m^3 s^{-1}$) above a threshold, for 2 time periods: discharges within the last 15 years (since 2001) vs. discharges during the entire period (1874-2015). The highest threshold is 50 years, and the lowest is 5 years. The discharge thresholds are based on return periods for 5, 10, 25 and 50 year-storms. For example, any storm with discharge higher than 3901 $m^3 s^{-1}$ is a 50-year storm.

| Threshold Discharge | Return Period (years) | Number of Discharges (2001-2015) | Number of Discharges (1874-2015) | % Discharges during the last 15 years |
|---|---|---|---|---|
| 3901 | 50 | 3 | 5 | 60 |
| 3566 | 25 | 7 | 20 | 35 |
| 3076 | 10 | 27 | 121 | 22 |
| 2653 | 5 | 49 | 249 | 20 |



Table 3. Minimum and maximum nutrient loads measured during the elevated storm discharge in June-July 2013 near Hamburg, Germany. The loads were reproduced with permission from the BSH report (Weigelt-Krenz et al. 2014).

| 12.06. - 8.07. 2013 | $NO_3$ (tons/day) | $NH_4$ (tons/day) | $PO_4$ (tons/day) | Si (tons/day) |
|---|---|---|---|---|
| **Min** | 200 | 5 | 3 | 400 |
| **Max** | 1100 | 28 | 15 | 1300 |
| **June average (1996-2005)** | 105 | na | 2.3 | 24 |



Table 4. Sampling dates of the different stations and platforms during different months in 2013.

| Source | March | July | August | September |
|---|---|---|---|---|
| **BSH** | 15-16 | 9-11 | 10-12 | 11-13 |
| **Helgoland Transects** | 25-27 | 2-4 | 6-8 | 4-5 |
| **FerryBox *M/V Funny Girl*** | none | 9-11 | 10-12 | 11-13 |
| **Deutsche Bucht (MARNET)** | 15-16 | none | 10-12 | 11-13 |





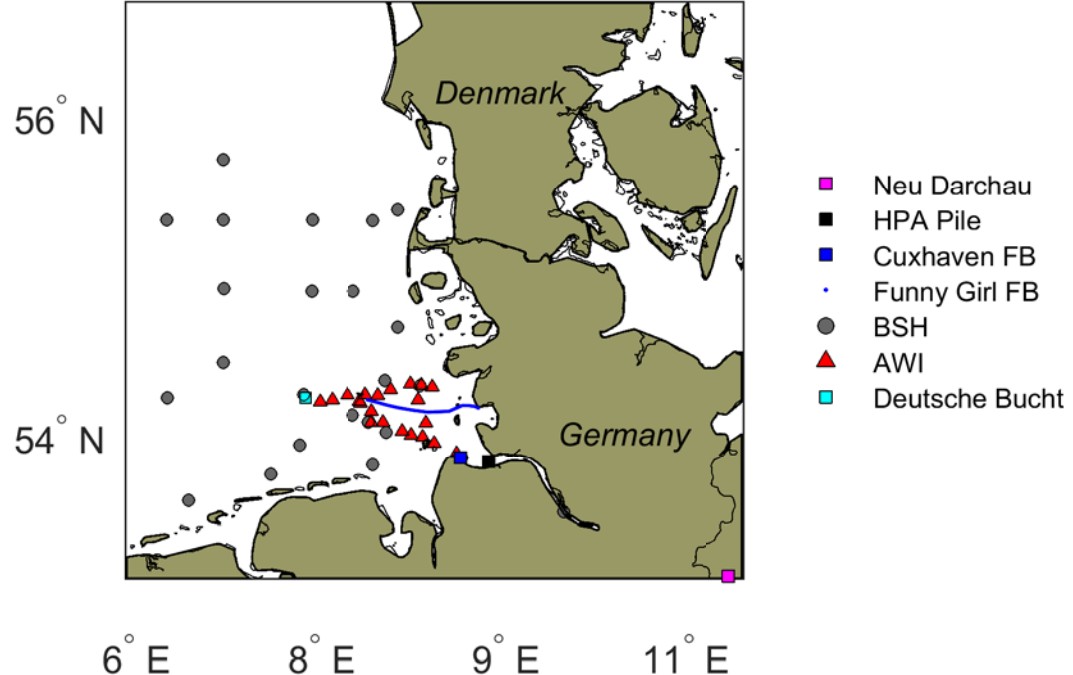

Fig. 1: Map of German Bight, Elbe Estuary and the continental areas around them. Stations are indicated with different symbols. Neu Darchau discharge gauging station (magenta square), operated by the German Federal Waterways and Shipping Administration (WSV); HPA Pile (black square), operated by BSH and HZG; Cuxhaven FerryBox (FB, blue square), operated by HZG, *M/V Funny Girl* FB transect (blue) between Büsum and Helgoland, operated by HZG; BSH discrete sampling stations (grey circles); AWI discrete sampling stations (red triangles); Deutsche Bucht MARNET monitoring station, operated by BSH (cyan square).





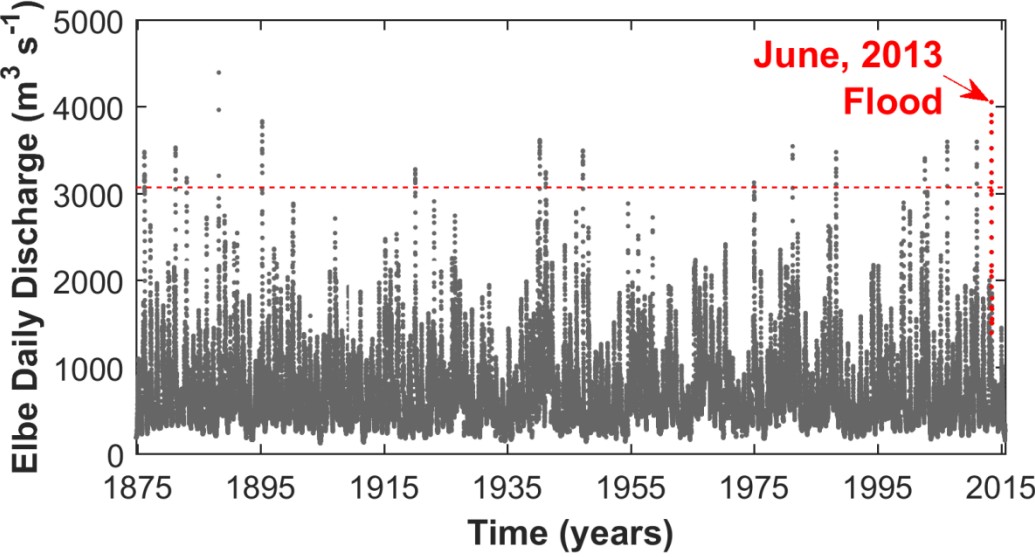

Fig. 2: Daily discharge from the Elbe River between 1874 and 2015. The dashed red line indicates the level of 10-year storm, as listed in Table 2. The June, 2013 flood discharge is highlighted in red and indicated with an arrow.





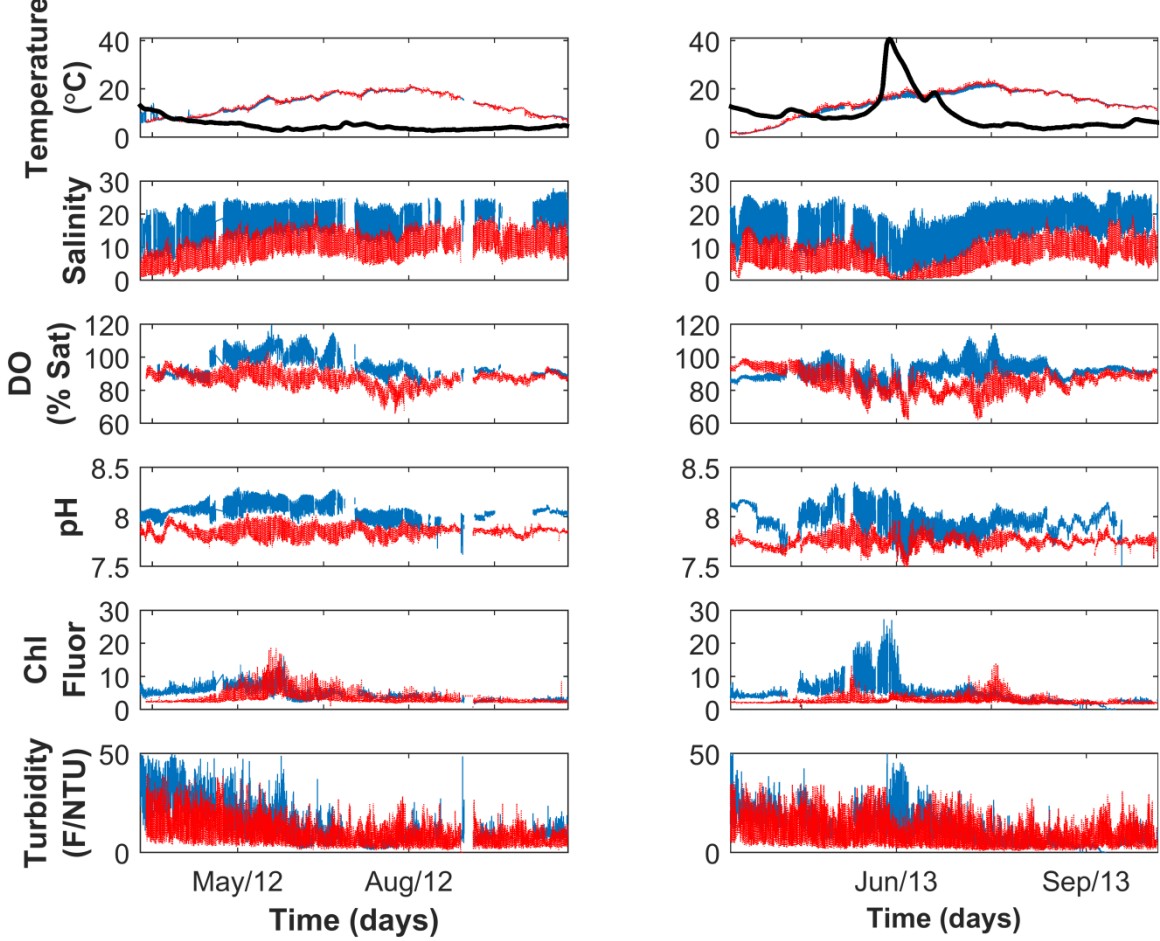

Fig. 3: Temperature, salinity, DO (% saturation), pH, chlorophyll (µg L-1), and turbidity (F/NTU), measured at Cuxhaven (725 river km, blue) and HPA Pile (710 river km, red dotted line) in the Elbe Estuary, for 2012 (left panels) and 2013 (right panels). As a reference, the Elbe discharge (m³ s⁻¹) at Neu Darchau station, scaled by dividing it by 100, was also included in the temperature plots.



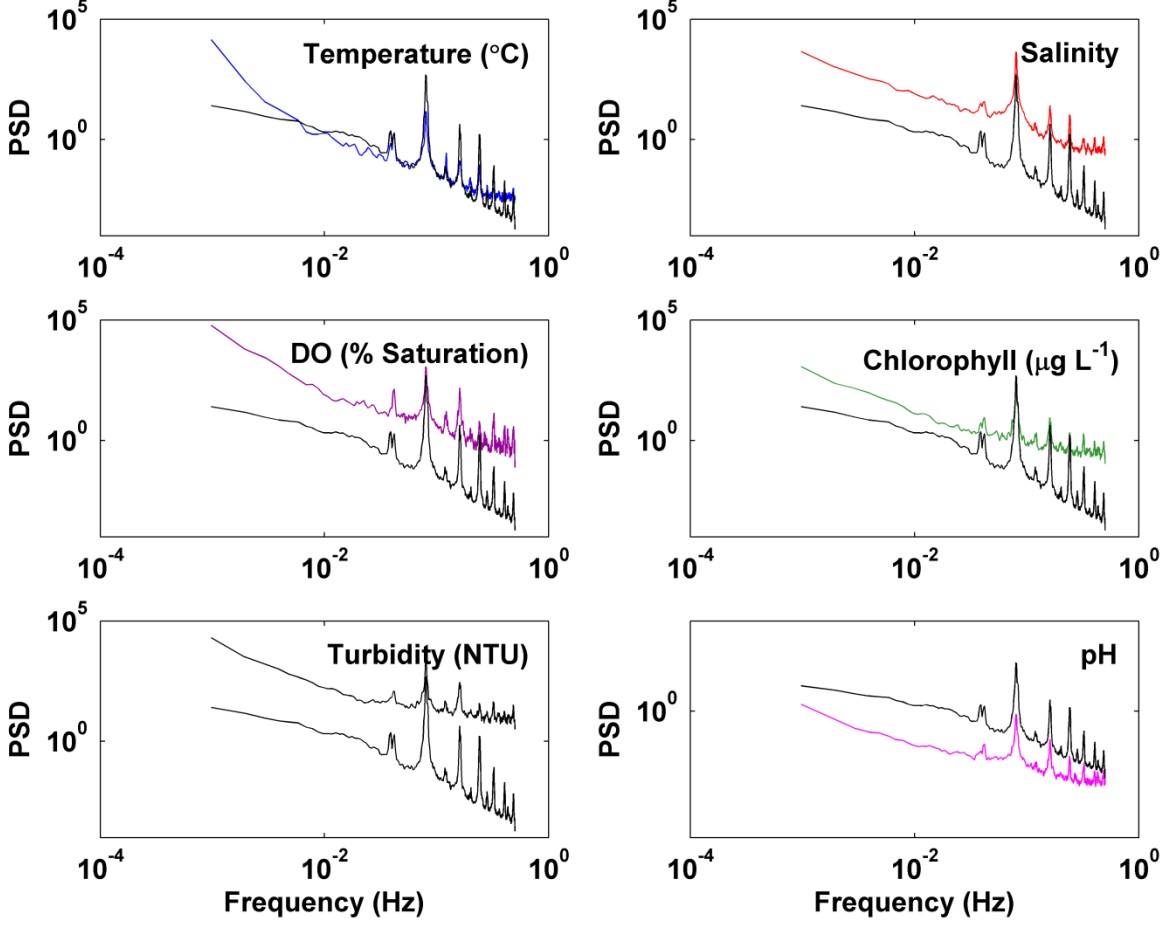

Fig. 4: Power spectral density (PSD) plots of 6 parameters measured at the Cuxhaven FerryBox station (temperature, salinity, DO, chlorophyll, turbidity and pH. Also shown on each panel in black is the PSD for sea level measured at a station near Cuxhaven.





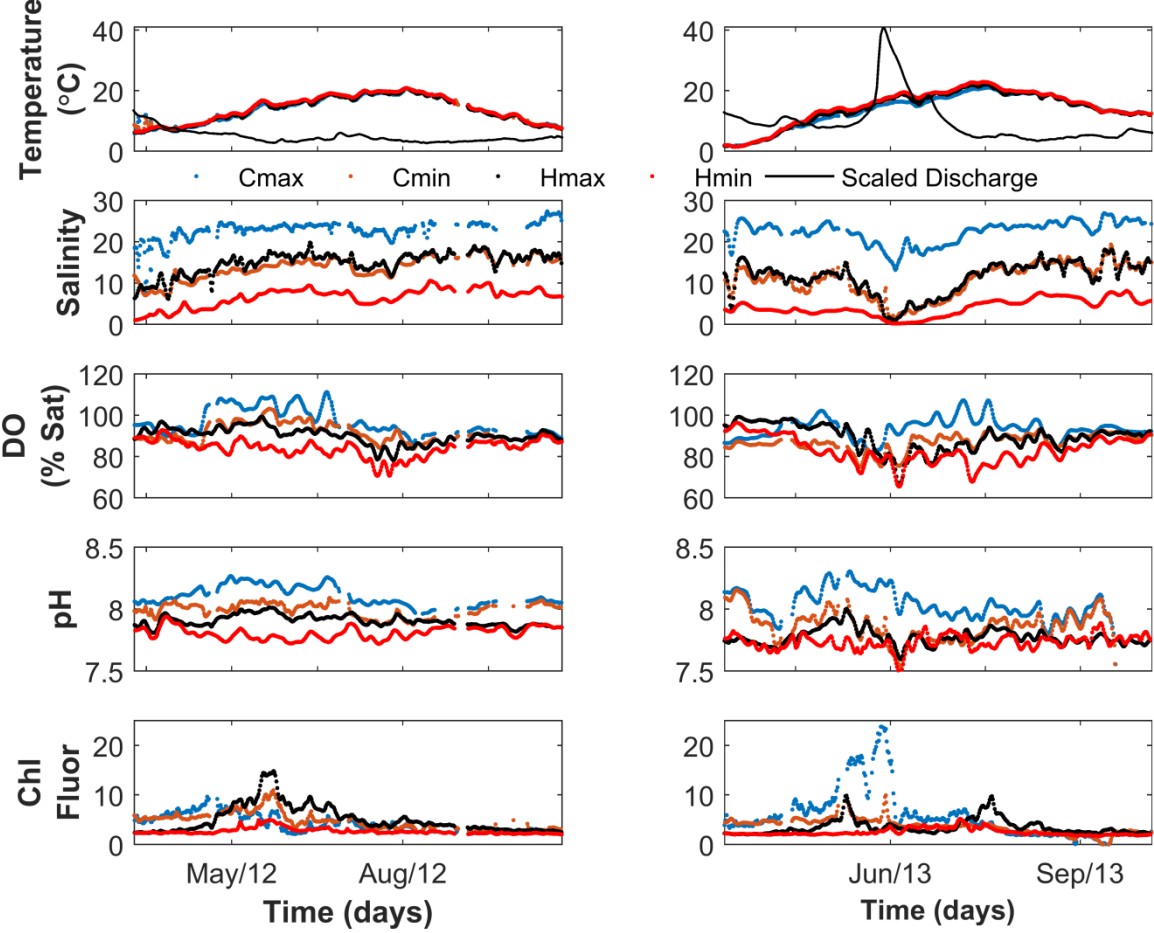

Fig. 5: Temperature, salinity, DO (% saturation), pH and chlorophyll (µg L$^{-1}$) measured at Cuxhaven and HPA Pile in the Elbe Estuary, for 2012 (left panels) and 2013 (right panels). The colors represent the data identified for each parameter, and at each station, at the times of salinity maxima (Cuxhaven, blue; HPA Pile, orange), and salinity minima (Cuxhaven, black; HPA Pile, red). As a reference, the Elbe discharge (m$^3$ s$^{-1}$) at Neu Darchau station, scaled by dividing it by 100 was also included in the temperature plots.





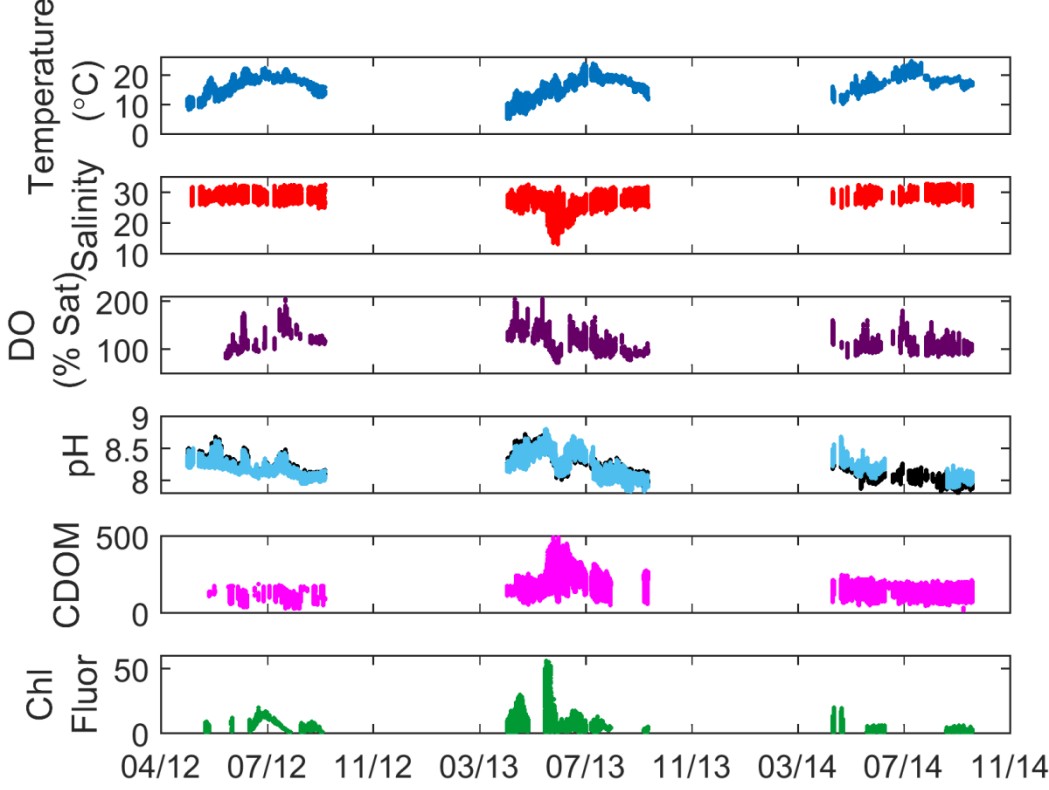

Fig. 6: FerryBox data from *M/V Funny Girl* for temperature, salinity, dissolved oxygen (% saturation), pH, CDOM, and chlorophyll fluorescence between May 2012, and October, 2015. For pH, there were 2 electrodes, one in black and the other in light blue.



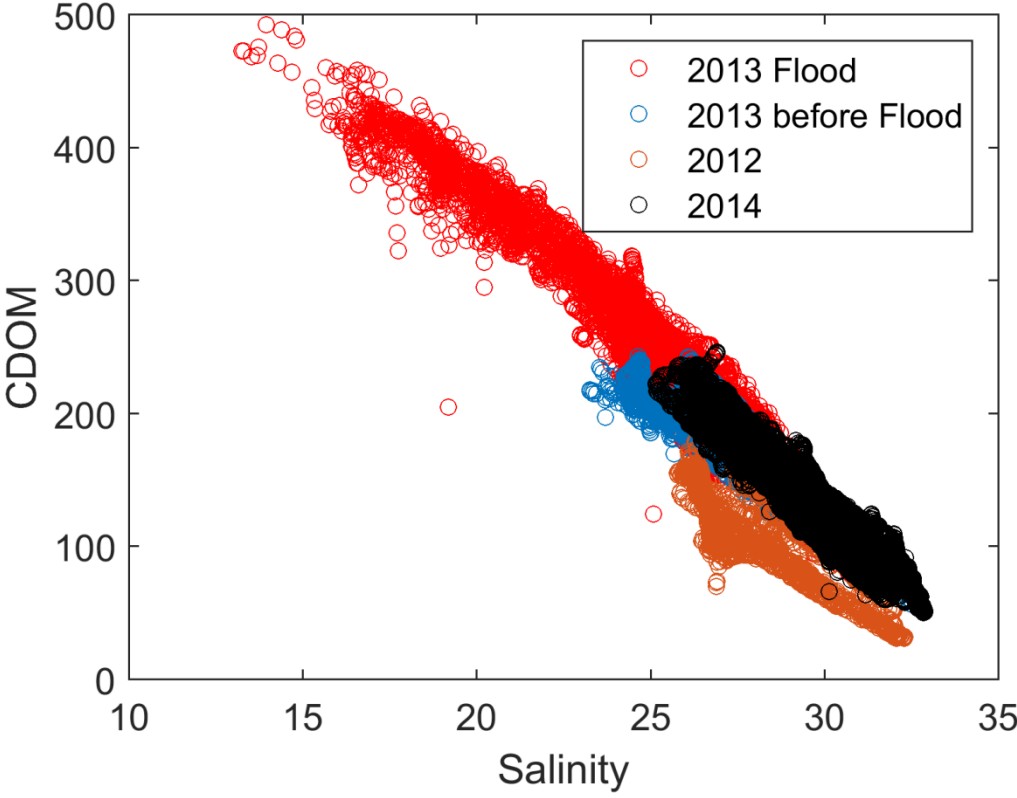

Fig. 7: Salinity vs. CDOM fluorescence from *M/V Funny Girl* for 2012 and 2014 summer seasons, as well as pre- and post- flood in 2013.





Fig. 8: Salinity (left) and temperature (right) between Büsum and Helgoland for each summer between 2008 and 2014, collected by the M/V Funny Girl FerryBox. The ferry transect is shown in Fig. 1.





Fig. 9: pH ferry data between Büsum and Helgoland for each summer between 2008 and 2014. There were two pH probes available on the ferry M/V Funny Girl. The white sections represent times when data were not available.





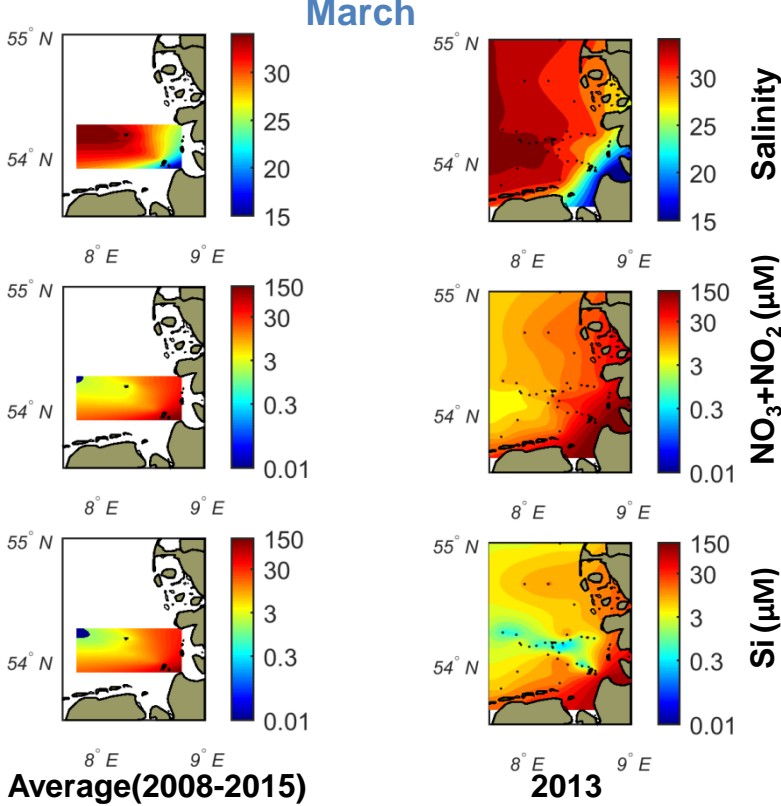

Fig. 10: Maps of interpolated (Kriging method of interpolation) salinity, nitrate + nitrite (NO₃+NO₂ (µM)), and silicate (Si (µM)) for the month of March. The left panels show average parameter distributions, based on 7 years of data (2008-2015, excluding 2013) from AWI stations (Table 1); the right panels show interpolated parameters from 2013, measured at AWI, BSH, FerryBox and HPA stations (Table 1 and 4), were sampled between 15 and 27 March, 2013.



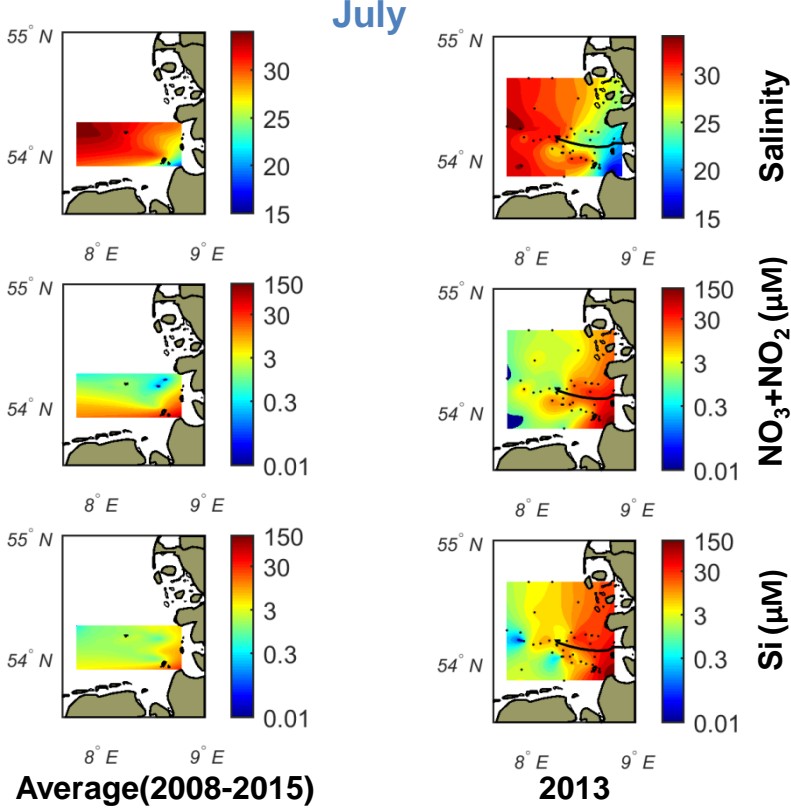

Fig. 11: Same description as in Fig. 10, for the month of July. The stations used in the right panel maps (Table 1 and 4) were sampled between 2 and 11 July, 2013.



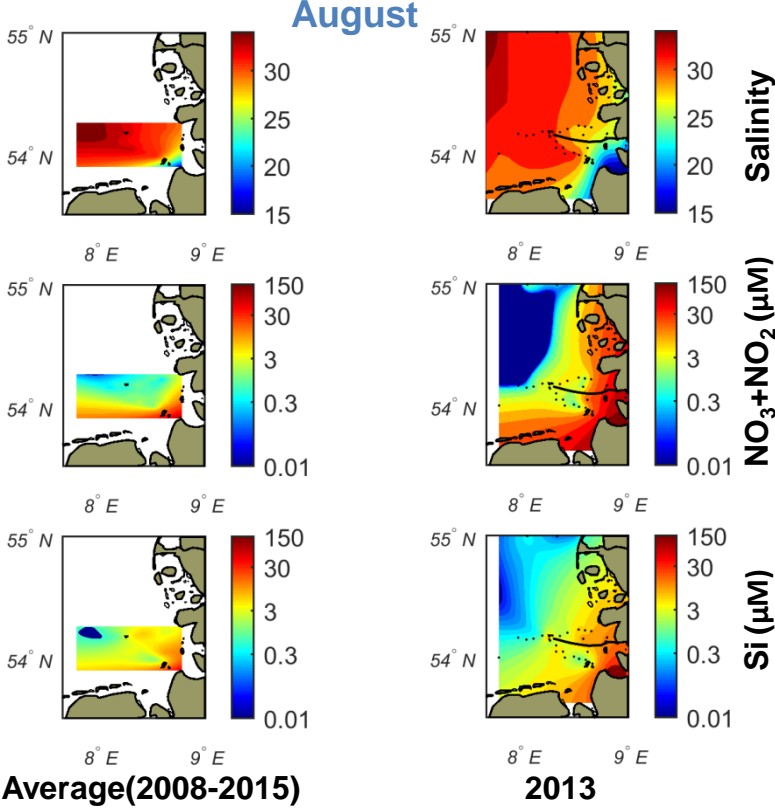

Fig. 12: Same description as in Fig. 10, for the month of August. The stations used in the right panel maps (Table 1 and 4) were sampled between 6 and 12 August, 2013.





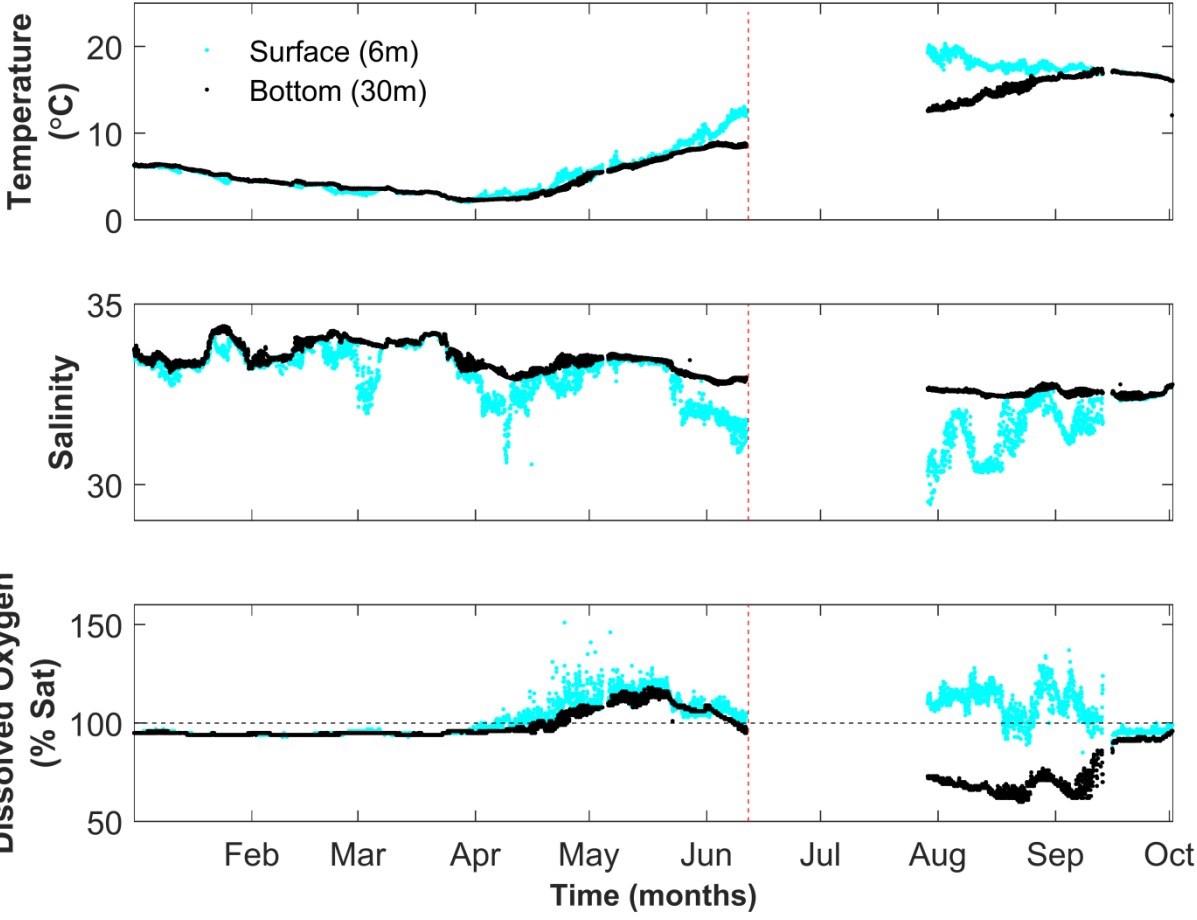

Fig. 13: Surface (6 m, cyan) and bottom (30 m, black) temperature, salinity and dissolved oxygen (% saturation) measured at the Deutsche Bucht station (Fig. 1, Table 1), part of the MARNET monitoring network. The data covers a time frame between January and October, 2013. The onset of the June flood is marked by a red vertical line.





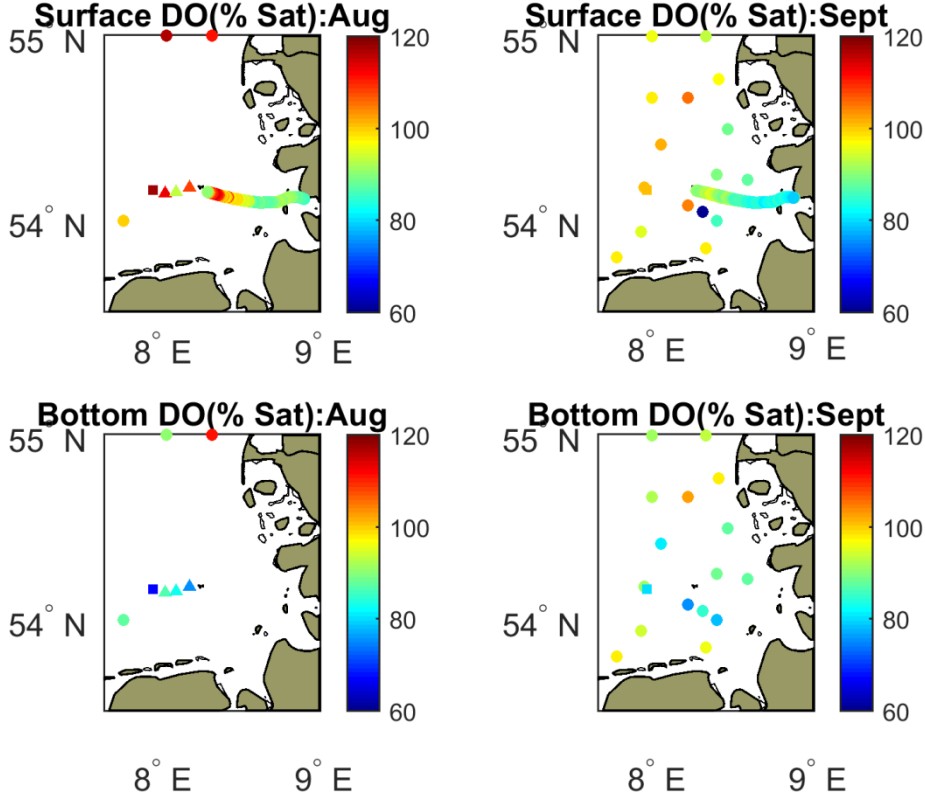

Fig. 14: Dissolved oxygen (% saturation) in surface and bottom waters measured in August and September. The surface and bottom dissolved oxygen were measured at available discrete stations from AWI and BSH stations, along with FerryBox (M/V Funny Girl) and
5   Deutsche Bucht MARNET station. The dates of coverage are listed in Table 4.