# Peer review of "Extreme Flood Impact on Estuarine and Coastal Biogeochemistry: the 2013 Elbe Flood"

_Biogeosciences, 2016_

## Referee Comment (RC1) · Anonymous Referee #1 · 11 Jul 2016

Review of Voynova et al. "Extreme flood impact on estuarine and coastal biogeochemistry: the 2013 Elbe flood"

This manuscript describes the effects of a large flood on coastal waters of the German Bight, with the broader relevance being that climate change is expected to lead to more frequent floods and thus it is important to understand their impact. This topic has been covered fairly extensively elsewhere for estuarine systems, but less so for the estuarine-coastal continuum. As such, it does add something new to the literature. I do have concerns that I hope the authors can address:

1) My main criticism has to do with the quality of the ferry-based data. For example, the authors discuss applying a correction to the pH data, which experienced drift. However, Figure S1 shows a much larger amount of drift than I was expecting. Quite honestly,

[Figure]

I do not feel that any form of drift correction will give me confidence in the usage of that data. I'd be inclined to tell the authors to remove the pH data altogether. Likewise, I believe that there needs to be more time spent on discussing the qa/qc procedures for the other ferry-based data, particularly D.O. and chlorophyll. The authors mention correcting the D.O. data. Did it experience drift of the magnitude that pH did? What were the procedures used to calibrate sensors (pre- and post-deployment)?

2) In regards to data presentation, I am perplexed as to why no discrete data is presented from the most critical time period, i.e., April-June 2013? It would really help to know what the conditions were like just prior to the flood.

3) The manuscript is generally well written and detailed in its analysis. I do feel as if it contains some analyses/text that are not fully relevant and could be removed to make the paper more succinct.

4) For example, much of the discussion on load calculations and residence time (bottom of page 12, all of page 13) seemed distracting from the overall message and rather drawn out. I suggest removing this section and just say somewhere that the residence time change from X to Y.

5) Much is made of an apparent deviation from linearity in the CDOM/salinity relationship (e.g., first paragraph page 15, third paragraph page 18). Yet I cannot see the non-linearity in Figure 7. Is it statistically significant?

6) Page 15, line 19- I think you mean Figure 9, not Figure 8.

7) The authors refer to loadings of particulate organic carbon throughout the ms, but did not actually measure this. Please rephrase or remove this, as it could lead a reader to believe that this was actually measured.

8) Figure 3 is fairly redundant with Figure 5.

9) I believe that the authors should spend more time on the longer-term effects of this flood, as that is a really exciting part of the story. For example, it appears to me that

the data in Figure 6 shows generally lower D.O and pH (even though I don't believe the pH data...) in 2014 compared to prior years. Could this be an artifact of the continued degradation of organic matter that entered the system during the 2013 flood? I believe that Paerl et al. 2001 observed these longer-term effects from a series of hurricanes.

10) Figure 8, x-axis labels longitude as "North". Should be East or West.

11) Figure 14, need to indicate year in Legend.

---

## Referee Comment (RC2) · Anonymous Referee #2 · 6 Aug 2016

GENERAL COMMENTS:

The paper by Voynova et al. aims at describing the effect of an extreme flood event occurred during summer of year 2013, on the biogeochemistry of the Elbe estuary and the adjacent coastal zone. The study demonstrates that a specific biogeochemical response of the Elbe system can be related to the large flood event by comparing biogeochemical data measured during three years (2012-2014) and confirms that the increasing frequency of this kind of phenomena in the future could strongly alter the current carbon cycle along the land-ocean aquatic continuum. Overall, the paper is well written and the work is detailed. However, I feel a number of revisions would be needed to improve its clarity. My recommendations are given below.

SPECIFIC COMMENTS:

[Figure]
SECTION 2.1: A map describing the study site should be added. This will help readers, who are not familiar with the Elbe coastal system, follow the study. Authors would also modify the existing Fig. 1. In this case, I would suggest specifying where the German Bight, the Wadden Sea, Zollenspieker and Geesthacht are located on the map. In both cases, I would also suggest specifying Büsum and Helgoland locations. Additionally, please specify in the caption of Fig. 1 what HPA, BSH, HZG and AWI stand for.

PAGE 5, lines 24-26: In Volta et al. (Volta et al., 2016. Regional carbon and CO2 budget of North Sea tidal estuaries, in: Estuarine, Coastal and Shelf Science), authors reported a very high average pH value ($\approx$9) at the Geesthacht weir during summer between years 2009-2011 and highlighted the uncertainty associated to this result. As a consequence, I am wondering if a pH>9.5, as reported in the manuscript, could be considered as a chronic condition for the riverine zone of the Elbe. Please clarify this aspect.

PAGE 7, lines 24-27: The drift of pH data is extremely large (Fig. S1). Please support the use of the method applied with literature. Mentioning if this method has been used before to correct biogeochemical drifted data will definitely strengthen the reliability of pH corrected data used in this study.

PAGE 9, line 16: Please specify which parameters influenced by biological production you focussed on.

PAGE 12, EQ. 1: Please explain better where this equation comes from (e.g. how many observed data have been used to obtain it?)

PAGE 14, line 4: It is unclear to me if data in Fig. 6 represent measurements in a specific location along the ferry transect or if they represent the average calculated over it. Please specify.

PAGE 14, lines 11-15: I would provide the R2 values relative to the linear correlations found between CDOM and salinity (Fig. 7). This would strengthen the result indicating

that there are no significant sinks/sources of DOC.

TECHNICAL COMMENTS:

PAGE 7, lines 25-26: The reference Aguilera, 2008 in the text is indicated as Aguilera, 2008b in the reference list. Please check.

PAGE 8, lines 18-19: Please remove "a" between "at" and "the surface".

PAGE 9, line 10: Please be coherent with the legend of Fig. 1. Does BAH AWI in the text correspond to AWI in Fig. 1's legend?

PAGE 13, lines 15-17: Please remove the " strong" referred to the linear correlation between TOC and salinity and between TSS and TOC.

PAGE 13, line 19: I think there is a missing "and" between "Cuxhaven" and "transport". Please check.

PAGE 14, line 27: I think authors mean Fig. 6, not 7. Please check.

PAGE 15, line 19: I think authors mean Fig. 9, not 8. Please check.

PAGE 19, line 19: I think authors mean Fig. 3, not 8. Please check.

PAGE 12-14: Please mention that these percentages refer to the Elbe.

FIG. 3: Please add a legend to specify what black, red and blue lines represent.

FIG. 4: Please add a legend to specify what black lines represent. Moreover, line colours for turbidity and sea level PSD are too similar. Please choose a different colour for turbidity.

FIG. 5: Please modify the legend and/or the caption to better explain what Cmax, Cmin, Hmax and Hmin stand for.

FIGS. 8 and 9: I think that x-axes should be labelled East (E) in both figures. Please check.

FIGS. 11 and 12: Please specify that black lines in the right panels represent the FB transect in both figures.

FIG. 14: Please mention that data refer to the year 2013 in the caption.

---

## Author Comment (AC1) · 9 Sep 2016

Dear reviewers,

Thank you for your comments, we appreciate your critical reviews and your help in trying to make this manuscript better. Please find our answers to reviewer comments below each comment.
* * *
Review of Voynova et al. "Extreme flood impact on estuarine and coastal biogeochemistry: the 2013 Elbe flood"

This manuscript describes the effects of a large flood on coastal waters of the German Bight, with the broader relevance being that climate change is expected to lead to

more frequent floods and thus it is important to understand their impact. This topic has been covered fairly extensively elsewhere for estuarine systems, but less so for the estuarine-coastal continuum. As such, it does add something new to the literature. I do have concerns that I hope the authors can address:

1) My main criticism has to do with the quality of the ferry-based data. For example, the authors discuss applying a correction to the pH data, which experienced drift. However, Figure S1 shows a much larger amount of drift than I was expecting. Quite honestly, I do not feel that any form of drift correction will give me confidence in the usage of that data. I'd be inclined to tell the authors to remove the pH data altogether. Likewise, I believe that there needs to be more time spent on discussing the qa/qc procedures for the other ferry-based data, particularly D.O. and chlorophyll. The authors mention correcting the D.O. data. Did it experience drift of the magnitude that pH did? What were the procedures used to calibrate sensors (pre- and post- deployment)? The drift in Fig. S1 was observed only in the pH data from a fixed station in the Elbe Estuary, the HPA Pile station. At this location, a pH electrode was deployed between March and November each year, and was not serviced and recalibrated during the time of deployment. Therefore, the electrode experienced a large amount of drift. The pH probes located on the Cuxhaven (fixed station) and MV Funny Girl FerryBox Systems were calibrated frequently (every time the probes were serviced, between weekly and monthly intervals). The pH probes located on the FerryBox Systems therefore did not experience drift. The drift correction of the HPA pile data was applied by removing the moving average for the yearly pH deployments. However, when compared to the calibrated pH record from the Cuxhaven FerryBox station, the drift corrected data showed the same average trend, therefore suggesting that the applied correction does not cause an atypical pH trend. However, since the drift cannot be verified against discrete samples and pH calibrations were not available to verify the record, we will remove the pH data from the HPA station from Figs. 3 and 5. Correction of the dissolved oxygen data on the Cuxhaven FerryBox was necessary because at this station, the FerryBox optode measured slightly lower oxygen levels than the Winkler Titration samples. However, the

offset is consistent throughout the record and no drift was observed. Only 4 Winkler samples (each in duplicate) were available for the correction of the DO time series at this station, however, the standard deviation for the duplicates was small. The optodes placed on the HPA Pile station were only serviced at the beginning and end of each seasonal deployment, and there were no Winkler samples collected to check the DO measurements; however, as no drift in the DO at this station was detected, this record was not corrected. We will include this information and will expand the discussion in the methods section on qa/qc of the data as suggested by the reviewer.

2) In regards to data presentation, I am perplexed as to why no discrete data is presented from the most critical time period, i.e., April-June 2013? It would really help to know what the conditions were like just prior to the flood. Unfortunately, there were no discrete data available for April-June, 2013. It would have been very useful to have discrete data for this period. The only data available for this late spring-early summer period came from the MV Funny Girl FerryBox.

3) The manuscript is generally well written and detailed in its analysis. I do feel as if it contains some analyses/text that are not fully relevant and could be removed to make the paper more succinct. We thank the reviewer for the overall assessment of our manuscript. We will try to correct the manuscript accordingly, and shorten or remove the section describing the discharge and residence time (section 3.3, particularly related to residence time equation calculation), and shorten the sections suggested by the reviewer below, which should improve the readability of the manuscript.

4) For example, much of the discussion on load calculations and residence time (bottom of page 12, all of page 13) seemed distracting from the overall message and rather drawn out. I suggest removing this section and just say somewhere that the residence time change from X to Y. We will shorten this section, remove equation (1), but keep the section about salinity response to discharge, and we will keep the paragraph about nutrient loads. Also, the section relating POC and chlorophyll based on FGG Elbe data will be shortened or removed.

5) Much is made of an apparent deviation from linearity in the CDOM/salinity relationship (e.g., first paragraph page 15, third paragraph page 18). Yet I cannot see the non-linearity in Figure 7. Is it statistically significant? We will include the linear relationships for these figures to show significance.

6) Page 15, line 19- I think you mean Figure 9, not Figure 8. Corrected.

7) The authors refer to loadings of particulate organic carbon throughout the ms, but did not actually measure this. Please rephrase or remove this, as it could lead a reader to believe that this was actually measured. This section will be removed from final manuscript in section 3.3, but some of the language relating POC to the flood will be clarified to specify that POC was not measured for this study specifically.

8) Figure 3 is fairly redundant with Figure 5. Whereas Figure 3 shows the general variability and data spread in the various biogeochemical parameters measured at two stations in the Elbe Estuary, Figure 5 focuses on the values measured at the minima and maxima salinity of each tidal cycle for each station. Figure 3 shows the raw data from the 2 stations for those readers who are interested to see the original data quality and comparison between the 2 stations in the estuary. Figure 5 is useful to show the variations occurring at each salinity (or water mass) end member, for each station, as well as the overlap between the 2 stations. It also shows with much more clarity that higher chlorophyll fluorescence was measured over the high salinity coastal area adjacent to the Elbe Estuary, compared to the lower chlorophyll fluorescence measured in the Estuary, between Cuxhaven and HPA pile stations. This suggests that a bloom existed in this coastal region. With Figure 5, we were able to also show the difference in water mass end members for other biogeochemical parameters sensitive to primary production and respiration, like dissolved oxygen and pH. Therefore, in our opinion, we should keep both of these figures in the manuscript.

9) I believe that the authors should spend more time on the longer-term effects of this flood, as that is a really exciting part of the story. For example, it appears to me that
the data in Figure 6 shows generally lower D.O and pH (even though I don't believe the pH data...) in 2014 compared to prior years. Could this be an artifact of the continued degradation of organic matter that entered the system during the 2013 flood? I believe that Paerl et al. 2001 observed these longer-term effects from a series of hurricanes. We thank the reviewer for this comment and suggestion. We will expand this section to the extent possible as the available data limits the extent to which we can discuss the long-term changes.

10) Figure 8, x-axis labels longitude as "North". Should be East or West. To be corrected.

11) Figure 14, need to indicate year in Legend. Corrected.

Review #2 SECTION 2.1: A map describing the study site should be added. This will help readers, who are not familiar with the Elbe coastal system, follow the study. Authors would also modify the existing Fig. 1. In this case, I would suggest specifying where the German Bight, the Wadden Sea, Zollenspieker and Geesthacht are located on the map. In both cases, I would also suggest specifying Büsum and Helgoland locations. Additionally, please specify in the caption of Fig. 1 what HPA, BSH, HZG and AWI stand for. We have clarified the map and added the suggested sites by the reviewer.

PAGE 5, lines 24-26: In Volta et al. (Volta et al., 2016. Regional carbon and CO2 budget of North Sea tidal estuaries, in: Estuarine, Coastal and Shelf Science), authors reported a very high average pH value ($\sim$9) at the Geesthacht weir during summer between years 2009-2011 and highlighted the uncertainty associated to this result. As a consequence, I am wondering if a pH>9.5, as reported in the manuscript, could be considered as a chronic condition for the riverine zone of the Elbe. Please clarify this aspect. Our data covers mostly the Elbe Estuary and German Bight coastal regions, and therefore we have limited ability to address this question, and clarify the pH. However, we will use other studies focused on the upper sections of the Elbe River, and

particularly the one near Geesthacht to better describe the pH in this region.

PAGE 7, lines 24-27: The drift of pH data is extremely large (Fig. S1). Please support the use of the method applied with literature. Mentioning if this method has been used before to correct biogeochemical drifted data will definitely strengthen the reliability of pH corrected data used in this study. This method was not applied based on another study. However, we will substantiate it based on other studies to the extent possible, or remove the pH data pertaining to HPA Pile, where this drift occurred, per suggestion of another reviewer.

PAGE 9, line 16: Please specify which parameters influenced by biological production you focused on. Corrected.

PAGE 12, EQ. 1: Please explain better where this equation comes from (e.g. how many observed data have been used to obtain it?) Upon suggestion by another reviewer we have decided to remove this section.

PAGE 14, line 4: It is unclear to me if data in Fig. 6 represent measurements in a specific location along the ferry transect or if they represent the average calculated over it. Please specify. The data from the entire ferry transect and their change in time are shown in Figure 6. In this way, we are able to show that after the June 2013 flood, the salinity and CDOM ranges along the ferry transect doubled, with the influx of lower salinity estuarine water. We will add explanation to the figure caption, and to the manuscript text to clarify this.

PAGE 14, lines 11-15: I would provide the R2 values relative to the linear correlations found between CDOM and salinity (Fig. 7). This would strengthen the result indicating that there are no significant sinks/sources of DOC. Thank you for this suggestion, the linear correlations will be included in order to strengthen the manuscript.

TECHNICAL COMMENTS: PAGE 7, lines 25-26: The reference Aguilera, 2008 in the text is indicated as Aguilera, 2008b in the reference list. Please check. Corrected.

PAGE 8, lines 18-19: Please remove "a" between "at" and "the surface". Corrected.

PAGE 9, line 10: Please be coherent with the legend of Fig. 1. Does BAH AWI in the text correspond to AWI in Fig. 1's legend? Corrected.

PAGE 13, lines 15-17: Please remove the "strong" referred to the linear correlation between TOC and salinity and between TSS and TOC. Corrected.

PAGE 13, line 19: I think there is a missing "and" between "Cuxhaven" and "transport". Please check. Corrected.

PAGE 14, line 27: I think authors mean Fig. 6, not 7. Please check. Corrected.

PAGE 15, line 19: I think authors mean Fig. 9, not 8. Please check. Corrected.

PAGE 19, line 19: I think authors mean Fig. 3, not 8. Please check. The reviewer probably is suggesting page 15, line 19 (there is no such reference on p.19, l.19), where the reference to Fig. 8 is correct. Fig. 8 is built from the moving FerryBox aboard the MV Funny Girl. This FerryBox had a long standing pH record since 2008, but only during the warm months between about April to September.

PAGE 12-14: Please mention that these percentages refer to the Elbe. We will shorten this section substantially, per suggestion of another reviewer, and will make sure to clarify it discusses the Elbe.

FIG. 3: Please add a legend to specify what black, red and blue lines represent. We added the description of the black line for discharge in the caption, and will add a legend.

FIG. 4: Please add a legend to specify what black lines represent. Moreover, line colours for turbidity and sea level PSD are too similar. Please choose a different colour for turbidity. This will be corrected.

FIG. 5: Please modify the legend and/or the caption to better explain what Cmax, Cmin, Hmax and Hmin stand for. Corrected by modifying the caption.

FIGS. 8 and 9: I think that x-axes should be labelled East (E) in both figures. Please check. This will be corrected.

[Figure]

*Dear reviewers,*

*Thank you for your comments, we appreciate your critical reviews and your help in trying to make this manuscript better. Please find our answers to reviewer comments below each comment.*
* * *
Review of Voynova et al. "Extreme flood impact on estuarine and coastal biogeochemistry: the 2013 Elbe flood"

This manuscript describes the effects of a large flood on coastal waters of the German Bight, with the broader relevance being that climate change is expected to lead to more frequent floods and thus it is important to understand their impact. This topic has been covered fairly extensively elsewhere for estuarine systems, but less so for the estuarine-coastal continuum. As such, it does add something new to the literature. I do have concerns that I hope the authors can address:

1) My main criticism has to do with the quality of the ferry-based data. For example, the authors discuss applying a correction to the pH data, which experienced drift. However, Figure S1 shows a much larger amount of drift than I was expecting. Quite honestly, I do not feel that any form of drift correction will give me confidence in the usage of that data. I'd be inclined to tell the authors to remove the pH data altogether. Likewise, I believe that there needs to be more time spent on discussing the qa/qc procedures for the other ferry-based data, particularly D.O. and chlorophyll. The authors mention correcting the D.O. data. Did it experience drift of the magnitude that pH did? What were the procedures used to calibrate sensors (pre- and post-deployment)?

*The drift in Fig. S1 was observed only in the pH data from a fixed station in the Elbe Estuary, the HPA Pile station. At this location, a pH electrode was deployed between March and November each year, and was not serviced and recalibrated during the time of deployment. Therefore, the electrode experienced a large amount of drift.*

*The pH probes located on the Cuxhaven (fixed station) and MV Funny Girl FerryBox Systems were calibrated frequently (every time the probes were serviced, between weekly and monthly intervals). The pH probes located on the FerryBox Systems therefore did not experience drift.*

*The drift correction of the HPA pile data was applied by removing the moving average for the yearly pH deployments. However, when compared to the calibrated pH record from the Cuxhaven FerryBox station, the drift corrected data showed the same average trend, therefore suggesting that the applied correction does not cause an atypical pH trend. However, since the drift cannot be verified against discrete samples and pH calibrations were not available to verify the record, we will remove the pH data from the HPA station from Figs. 3 and 5.*

*Correction of the dissolved oxygen data on the Cuxhaven FerryBox was necessary because at this station, the FerryBox optode measured slightly lower oxygen levels than the Winkler Titration samples. However, the offset is consistent throughout the record and no drift was observed. Only 4 Winkler samples (each in duplicate) were available for the correction of the DO time series at this station, however, the standard deviation for the duplicates was small. The optodes placed on the HPA Pile station were only serviced at the beginning and end of each seasonal deployment, and there were no Winkler samples collected to check the DO*

**Fig. 1.**

---

## Author Response (AR1)

Dear Editors:

This manuscript is a revision of the manuscript BG-2016-218, submitted as part of a **special issue** in Biogeosciences, titled *COSYNA: integrating observations and modeling to understand coastal systems (OS/BG inter-journal SI)*. This submission is intended as a Research Article, and contains only original data. If the manuscript is accepted for publication, any publication charges associated with it will be paid.

This manuscript, titled *Extreme Flood Impact on Estuarine and Coastal Biogeochemistry: the 2013 Elbe Flood*, examines the influence of an extreme Elbe River discharge event that took place in June 2013 on the biogeochemistry in the Elbe Estuary and the adjacent German Bight. The flood was the largest summer discharge on record within the last 140 years. The high-frequency monitoring network in the German Bight available within the Coastal Observing System for Northern and Arctic Seas (COSYNA) captured the flood influence on the German Bight. Monitoring data from a FerryBox station in the Elbe Estuary (Cuxhaven) and from a FerryBox platform aboard the *M/V Funny Girl Ferry* (travelling between Büsum and Helgoland) documented the salinity changes on the German Bight, which persisted for about 2 months after the peak discharge. The flood generated a large influx of low salinity plume water enriched in nutrients, dissolved and particulate organic carbon on the coast. Water column stratification, and high nutrient influx subsequently led to the onset of a chlorophyll bloom within the surface waters of the southeast German Bight, observed by dissolved oxygen supersaturation, and higher than usual pH. Prolonged stratification also led to widespread bottom water dissolved oxygen depletion, unusual for the southeast German Bight in the summer. From a discharge analysis of the 140-year daily discharge record, we found that the frequency of large to extreme discharge events has increased in recent years, and 20-60% of these discharges have occurred in the last 15 years. This suggests that these extreme events are becoming more frequent, and they can have substantial influence on the biogeochemistry in coastal and estuarine ecosystems.

We have detailed our responses the reviewer comments in a separate document (updated). Some of the major changes include removing two figures from the supplementary online materials, and adding one, revising the map (Fig. 1), and revising the salinity vs CDOM plot, per reviewer suggestions. We have removed a section discussing residence time and loads, and reorganized the manuscript slightly, which required a number of structural edits. We are not including the track changes of these edits.

Thank you,

Corresponding author:

Yoana G. Voynova
Helmholz-Zentrum Geesthacht
Institute of Coastal Research
Max Planck Str. 1
21502 Geesthacht, Germany
+49 4152 872377
yoana.voynova@hzg.de

*Dear reviewers,*

*Thank you for your comments, we appreciate your critical reviews and your help in trying to make this manuscript stronger. Please find our answers to reviewer comments below each comment.*
* * *
Review of Voynova et al. "Extreme flood impact on estuarine and coastal biogeochemistry: the 2013 Elbe flood"

This manuscript describes the effects of a large flood on coastal waters of the German Bight, with the broader relevance being that climate change is expected to lead to more frequent floods and thus it is important to understand their impact. This topic has been covered fairly extensively elsewhere for estuarine systems, but less so for the estuarine-coastal continuum. As such, it does add something new to the literature. I do have concerns that I hope the authors can address:

1) **My main criticism has to do with the quality of the ferry-based data. For example, the authors discuss applying a correction to the pH data, which experienced drift. However, Figure S1 shows a much larger amount of drift than I was expecting. Quite honestly, I do not feel that any form of drift correction will give me confidence in the usage of that data. I'd be inclined to tell the authors to remove the pH data altogether. Likewise, I believe that there needs to be more time spent on discussing the qa/qc procedures for the other ferry-based data, particularly D.O. and chlorophyll. The authors mention correcting the D.O. data. Did it experience drift of the magnitude that pH did? What were the procedures used to calibrate sensors (pre- and post- deployment)?**

*The drift in Fig. S1 was observed only in the pH data from a fixed station in the Elbe Estuary, the HPA Pile station. At this location, a pH electrode was deployed between March and November each year, and was not serviced and recalibrated during the time of deployment. Therefore, the electrode experienced a large amount of drift.*

*The pH probes located on the Cuxhaven (fixed station) and MV Funny Girl FerryBox Systems were calibrated frequently (every time the probes were serviced, between weekly and monthly intervals). The pH probes located on the FerryBox Systems therefore did not experience drift.*

*The drift correction of the HPA pile data was applied by removing the moving average for the yearly pH deployments. However, when compared to the calibrated pH record from the Cuxhaven FerryBox station, the drift corrected data showed the same average trend, therefore suggesting that the applied correction does not cause an atypical pH trend. However, since the drift cannot be verified against discrete samples and pH calibrations were not available to verify the record, we will remove the pH data from the HPA station from Figs. 3 and 5.*

*Correction of the dissolved oxygen data on the Cuxhaven FerryBox was necessary because at this station, the FerryBox optode measured slightly lower oxygen levels than the Winkler Titration samples. However, the offset is consistent throughout the record and no drift was observed. Only 4 Winkler samples (each in duplicate) were available for the correction of the DO time series at this station, however, the standard deviation for the duplicates was small. The optodes placed on the HPA Pile station were only serviced at the beginning and end of each seasonal deployment, and there were no Winkler samples collected to check the DO*

*measurements; however, as no drift in the DO at this station was detected, this record was not corrected. We have included this information and have expanded the discussion in the methods section on qa/qc of the data as suggested by the reviewer.*

2) **In regards to data presentation, I am perplexed as to why no discrete data is presented from the most critical time period, i.e., April-June 2013? It would really help to know what the conditions were like just prior to the flood.**

*Unfortunately, there were no discrete data available for April-June, 2013 from BSH, and so surface maps for these months were not generated. The only data available for this late spring-early summer period came from the MV Funny Girl FerryBox and from BAH AWI. We have included the discrete data of several parameters, salinity, nitrate + nitrite, silicate and phosphate in the supplementary online materials to show the changes in these parameters with longitude from June (before the storm) until August.*

3) **The manuscript is generally well written and detailed in its analysis. I do feel as if it contains some analyses/text that are not fully relevant and could be removed to make the paper more succinct.**

*We thank the reviewer for the overall assessment of our manuscript. We have tried to correct the manuscript accordingly, and we have removed the section describing the discharge and residence time (section 3.3, particularly related to residence time equation calculation), and shortened the sections suggested by the reviewer below. This has indeed improved the readability of the manuscript.*

4) **For example, much of the discussion on load calculations and residence time (bottom of page 12, all of page 13) seemed distracting from the overall message and rather drawn out. I suggest removing this section and just say somewhere that the residence time change from X to Y.**

*We have shortened this section, removed equation (1), but kept the section about salinity response to discharge, and about nutrient loads. Also, the section relating POC and chlorophyll based on FGG Elbe data has been removed.*

5) **Much is made of an apparent deviation from linearity in the CDOM/salinity relationship (e.g., first paragraph page 15, third paragraph page 18). Yet I cannot see the non-linearity in Figure 7. Is it statistically significant?**

*We have included the linear relationships for these figures to show linear regressions and equations. Even though visually it looks like CDOM during and after the flood deviated from linearity, the regression analysis proved this was not the case. Still, the large number of points may be overwhelming this signal, especially if it is happening in a specific area along the ferry transect. To keep things simple we have not focused on this. However, the change in slope of the salinity vs CDOM regressions after the flood has been suggested to show a change in the organic matter content in the coastal areas.*

6) **Page 15, line 19- I think you mean Figure 9, not Figure 8.**

*Corrected.*

7) **The authors refer to loadings of particulate organic carbon throughout the ms, but did not actually measure this. Please rephrase or remove this, as it could lead a reader to believe that this was actually measured.**

*This section has been removed from final manuscript in section 3.3, but some of the language relating POC to the flood had been clarified to specify that POC was not measured for this study specifically.*

8) **Figure 3 is fairly redundant with Figure 5.**

*Whereas Figure 3 shows the general variability and data spread in the various biogeochemical parameters measured at two stations in the Elbe Estuary, Figure 5 focuses on the values measured at the minima and maxima salinity of each tidal cycle for each station. Figure 3 shows the raw data from the 2 stations for those readers who are interested to see the original data quality and comparison between the 2 stations in the estuary. Figure 5 is useful to show the variations occurring at each salinity (or water mass) end member, for each station, as well as the overlap between the 2 stations. It also shows with much more clarity that higher chlorophyll fluorescence was measured over the high salinity coastal area adjacent to the Elbe Estuary, compared to the lower chlorophyll fluorescence measured in the Estuary, between Cuxhaven and HPA pile stations. This suggests that a bloom existed in this coastal region. With Figure 5, we were able to also show the difference in water mass end members for other biogeochemical parameters sensitive to primary production and respiration, like dissolved oxygen and pH. Therefore, in our opinion, we should keep both of these figures in the manuscript.*

9) **I believe that the authors should spend more time on the longer-term effects of this flood, as that is a really exciting part of the story. For example, it appears to me that the data in Figure 6 shows generally lower D.O and pH (even though I don't believe the pH data...) in 2014 compared to prior years. Could this be an artifact of the continued degradation of organic matter that entered the system during the 2013 flood? I believe that Paerl et al. 2001 observed these longer-term effects from a series of hurricanes.**

*We thank the reviewer for this comment and suggestion. We have expanded this section to the extent possible as the available data limited our discussion on the long-term changes. We have added sections on this in the discussion and conclusion sections of the paper.*

10) **Figure 8, x-axis labels longitude as "North". Should be East or West.**

*Corrected.*

11) **Figure 14, need to indicate year in Legend.**

*Corrected.*

Review #2
**SECTION 2.1: A map describing the study site should be added. This will help readers, who are not familiar with the Elbe coastal system, follow the study. Authors would also modify the existing Fig. 1. In this case, I would suggest specifying where the German Bight, the Wadden Sea, Zollenspieker and Geesthacht are located on the map. In both cases, I would also suggest specifying Büsum and Helgoland locations. Additionally, please specify in the caption of Fig. 1 what HPA, BSH, HZG and AWI stand for.**

*We have clarified the map and added the suggested sites by the reviewer. This has improved the map considerably.*

**PAGE 5, lines 24-26: In Volta et al. (Volta et al., 2016. Regional carbon and CO2 budget of North Sea tidal estuaries, in: Estuarine, Coastal and Shelf Science), authors reported a very high average pH value (~9) at the Geesthacht weir during summer between years 2009-2011 and highlighted the uncertainty associated to this result. As a consequence, I am wondering if a pH>9.5, as reported in the manuscript, could be considered as a chronic condition for the riverine zone of the Elbe. Please clarify this aspect.**

*Our data covers mostly the Elbe Estuary and German Bight coastal regions, and therefore we have limited ability to address this question, and clarify the pH. The high pH (>9.5) mentioned was taken from previous studies, as cited Petersen et al. 1999. However, this is an interesting question which could be addressed in future studies on the Elbe River Estuary.*

**PAGE 7, lines 24-27: The drift of pH data is extremely large (Fig. S1). Please support the use of the method applied with literature. Mentioning if this method has been used before to correct biogeochemical drifted data will definitely strengthen the reliability of pH corrected data used in this study.**

*This method was not applied based on another study. However, we have now removed the pH data pertaining to HPA Pile, where this drift occurred, per suggestion of another reviewer.*

**PAGE 9, line 16: Please specify which parameters influenced by biological production you focused on.**

*Corrected.*

**PAGE 12, EQ. 1: Please explain better where this equation comes from (e.g. how many observed data have been used to obtain it?)**

*Upon suggestion by another reviewer we have decided to remove this section, but keep the discussion on residence time based on the salinity changes following the river discharge.*

**PAGE 14, line 4: It is unclear to me if data in Fig. 6 represent measurements in a specific location along the ferry transect or if they represent the average calculated over it. Please specify.**

*The data from the entire ferry transect and their change in time are shown in Figure 6. In this way, we were able to show that after the June 2013 flood, the salinity and CDOM ranges along the ferry transect doubled, with the influx of lower salinity estuarine water. We have added explanation to the figure caption, and to the manuscript text to clarify this.*

**PAGE 14, lines 11-15: I would provide the R2 values relative to the linear correlations found between CDOM and salinity (Fig. 7). This would strengthen the result indicating that there are no significant sinks/sources of DOC.**

*We thank the reviewer for this suggestion: the linear regressions have been included, and the CDOM analysis has been altered to include a discussion on the slopes of the salinity vs. CDOM curves. In 2013, after the flood, the regression has an $R^2$ of 0.95, which suggests that there was no deviation from linearity, even if visually it seems like there is a deviation. However, we found that after the flood and in 2014, the slope of the salinity vs CDOM regression became*

*more negative than before the flood. We are using therefore the slopes to suggest that there was a prolonged influence of the flood on the carbon cycle in the German Bight.*

**TECHNICAL COMMENTS:**
**PAGE 7, lines 25-26: The reference Aguilera, 2008 in the text is indicated as Aguilera, 2008b in the reference list. Please check.**
    *Corrected. Reference was subsequently removed, as its corresonding section was also removed from the manuscript.*

**PAGE 8, lines 18-19: Please remove "a" between "at" and "the surface".**
    *Corrected.*

**PAGE 9, line 10: Please be coherent with the legend of Fig. 1. Does BAH AWI in the text correspond to AWI in Fig. 1's legend?**
    *Corrected.*

**PAGE 13, lines 15-17: Please remove the "strong" referred to the linear correlation between TOC and salinity and between TSS and TOC.**
    *Corrected.*

**PAGE 13, line 19: I think there is a missing "and" between "Cuxhaven" and "transport". Please check.**
    *Corrected.*

**PAGE 14, line 27: I think authors mean Fig. 6, not 7. Please check.**
    *Corrected.*

**PAGE 15, line 19: I think authors mean Fig. 9, not 8. Please check.**
    *Corrected.*

**PAGE 19, line 19: I think authors mean Fig. 3, not 8. Please check.**
    *The reviewer probably is suggesting page 15, line 19 (there is no such reference on p.19, l.19), where the reference to Fig. 8 is correct. Fig. 8 is built from the moving FerryBox aboard the MV Funny Girl. This FerryBox had a long standing pH record since 2008, but only during the warm months between about April to September.*

**PAGE 12-14: Please mention that these percentages refer to the Elbe.**
    *We have shortened this section substantially, per suggestion of another reviewer, and have revised accordingly.*

**FIG. 3: Please add a legend to specify what black, red and blue lines represent.**
    *We added the description of the black line for discharge in the caption.*

**FIG. 4: Please add a legend to specify what black lines represent. Moreover, line**

**colours for turbidity and sea level PSD are too similar. Please choose a different colour for turbidity**

*Corrected.*

**FIG. 5: Please modify the legend and/or the caption to better explain what Cmax, Cmin, Hmax and Hmin stand for.**

*Corrected by modifying the caption.*

**FIGS. 8 and 9: I think that x-axes should be labelled East (E) in both figures. Please check.**

*Corrected.*

---

## Editor Decision (ED1)

bg-2016-218

Extreme Flood Impact on Estuarine and Coastal Biogeochemistry: the 2013 Elbe Flood
Yoana G. Voynova, Holger Brix, Wilhelm Petersen, Sieglinde Weigelt-Krenz, and Mirco Scharfe

Dear Dr Voynova and co-authors,

Thank you for sending your revised manuscript version.
It is really unfortunate that you haven't kept track of the changes you have made to your text, and the task of comparing both previous and present versions is quite difficult. I strongly suggest you in the future to keep track of your manuscript versions in order to highlight the changes you might need to do.

Please refer to the attached comments, before the final decision on your manuscript.
Best wishes
Leticia Cotrim da Cunha
* * *
a) I have a question concerning the HPA Pile station, and the pH values (and other parameters depicted in Figure 5.
Your figure caption says:

*"Fig. 5: Temperature, salinity, DO (% saturation), pH and chlorophyll (µg L-1) measured at Cuxhaven and HPA Pile in the Elbe Estuary, for 2012 (left panels) and 2013 (right panels). The colors represent the data identified for each parameter, and at each station, at the times of salinity maxima (Cmax at Cuxhaven, blue; **Hmax at HPA Pile, orange**), and salinity minima (Cmin at Cuxhaven, black; **Hmin at HPA Pile, red**). As a reference, the Elbe River discharge (originally measured in m3 s-1) at Neu Darchau station (Fig. 1), was scaled by dividing it by 100 and was included in the temperature plots."*

I am really confused on the colour legend (for each panel – is it the same?), and especially for the panel showing pH. You said in the reply document that pH data from HPA Pile Station had been removed from figures 3 (this is fine) and 5, but I see an orange line in the pH panel here. Could you please revise this figure, and re-write the figure caption, so that you avoid any misunderstanding?

b) Now concerning Figures 10, 11, 12:
Fig. 10 caption says:

*"Fig. 10: Maps of interpolated (Kriging method of interpolation) salinity, nitrate + nitrite (NO3+NO2 (µM)), and silicate (Si (µM)) for the month of March. The left panels show average parameter distributions, based on 7 years of data (2008-2015, excluding 2013) from AWI stations (Table 1); the right panels show interpolated parameters from 2013, measured at AWI, BSH, FerryBox and HPA stations (Table 1 and 4), were sampled between 15 and 27 March, 2013."*

I'd suggest you to re-write the explanation about the right-hand panels. For instance: the right panels show interpolated parameters from March 15th to 27th , 2013, measured at AWI, BSH, FerryBox and HPA Stations (Tables 1 and 4). The same applies to figures 11 and 12.

c) Figure 14 caption says:

"Fig. 14: Dissolved oxygen (% saturation) in surface and bottom waters measured in August and September, ***2013meteorologicalmeteoro.*** The surface and bottom dissolved oxygen were measured at available discrete stations from AWI and BSH stations, along with FerryBox (M/V Funny Girl) and Deutsche Bucht MARNET station. The dates of coverage are listed in Table 4."

Maybe you forgot a bit of text in the caption (marked in bold, italic, underlined)?

---

## Author Response (AR2)

Dear Editor,

Thank you for your notes, please find the marked corrections in the latest version of the manuscript below. We will keep in mind your recommendations for future submissions, and we appreciate your
5 help with reviewing our manuscript despite the difficulties without the tracked changes.

Please let us know if you need any further documents.

Best regards,
10
Yoana G. Voynova

[revised manuscript text omitted]